# Precision measurements on oxygen formation in stellar helium burning with gamma-ray beams and a Time Projection Chamber

R. Smith [1,2✉], M. Gai [2], S. R. Stern [2], D. K. Schweitzer [2] & M. W. Ahmed[3,4]

The carbon/oxygen (C/O) ratio at the end of stellar helium burning is the single most important nuclear input to stellar evolution theory. However, it is not known with sufficient accuracy, due to large uncertainties in the cross-section for the fusion of helium with $^{12}$C to form $^{16}$O, denoted as $^{12}$C($\alpha, \gamma$)$^{16}$O. Here we present results based on a method that is significantly different from the experimental efforts of the past four decades. With data measured inside one detector and with vanishingly small background, angular distributions of the $^{12}$C($\alpha, \gamma$)$^{16}$O reaction were obtained by measuring the inverse $^{16}$O($\gamma, \alpha$)$^{12}$C reaction with gamma-beams and a Time Projection Chamber (TPC) detector. We agree with current world data for the total reaction cross-section and further evidence the strength of our method with accurate angular distributions measured over the 1$^-$ resonance at $E_{cm} \sim 2.4$ MeV. Our technique promises to yield results that will surpass the quality of the currently available data.

[1] Department of Engineering and Mathematics, Sheffield Hallam University, Sheffield S1 1WB, UK. [2] Laboratory for Nuclear Science at Avery Point, University of Connecticut, Groton, CT 06340-6097, USA. [3] Triangle Universities Nuclear Laboratory, Duke University, Durham, NC 27708-0308, USA. [4] Department of Mathematics and Physics, North Carolina Central University, Durham, NC 27707, USA. ✉email: robin.smith@shu.ac.uk

Nuclear astrophysics, the study of nuclear processes in stars and in the cosmos, is a mature science that, among other topics, led to the theory of stellar evolution. In this theory, the uncertainty of the C/O ratio at the end of helium burning still remains significant. In his Nobel speech in 1984, W.A. Fowler stated "the ratio of $^{12}$C to $^{16}$O in helium burning is of paramount importance in nuclear astrophysics"[1], as it still is today. For example, it determines the fate of Type II supernovae (black hole or neutron star) as well as the light curves of Type Ia supernovae, which are used to measure cosmological distances, leading to the recent discovery of the accelerated expansion and Dark Energy.

In stellar helium burning, carbon and oxygen are formed. Since the formation of carbon is rather well understood, the C/O ratio is determined largely by the single remaining uncertain process: the fusion of $^{12}$C with an alpha-particle to form $^{16}$O, denoted as the $^{12}$C$(\alpha, \gamma)^{16}$O reaction. This reaction was measured in terrestrial laboratories to energies below 2 MeV, but it needs to be extrapolated to stellar conditions of a plasma with a temperature of $kT \sim 20$ keV, and the most efficient burning energy of 300 keV (the Gamow window[1]). In stellar conditions, two partial waves, $\ell = 1$ and 2, contribute, and they are denoted by the spectroscopic $E1$ and $E2$ amplitudes in reaction and scattering theory[1]. The challenges in this field are measurements of angular distributions of the $^{12}$C$(\alpha, \gamma)$ reaction from which the $E1$ and $E2$ cross sections are extracted, for accurate extrapolations to stellar conditions.

Progress was achieved in measuring angular distributions of the $^{12}$C$(\alpha, \gamma)$ reaction by directly measuring the emitted gamma-rays[2–9]. However, large uncertainties remain in the measured $E1$ and $E2$ cross sections and their extrapolations to stellar conditions at the Gamow window, $S_{E1}(300)$ and $S_{E2}(300)$, respectively[1]. In the latest extrapolation to stellar conditions using R-matrix analysis, deBoer et al.[10] examined the current world data and concluded that a "level of uncertainty ~10% may be in sight".

Previously measured angular distributions[2–9] were fitted with the three parameters: $E1$ and $E2$ amplitudes and their mixing phase angle ($\phi_{12}$), as outlined by Dyer and Barnes[2,11]. The extracted $\phi_{12}$ values at energies of $E_{cm} < 2.0$ MeV[8,9] were found in agreement with the theoretical prediction[2,11]: $\phi_{12} = \delta_2 - \delta_1 + \tan^{-1} \eta/2$, where $\delta_1$ and $\delta_2$ are the measured $\alpha + ^{12}$C elastic scattering phase shifts, and for this system, $\eta = 12 \times \alpha/\beta$, where $\alpha$ is the fine structure constant and $\beta = v/c$. In the energy region immediately below 2.0 MeV, the ratio $E1/E2$ and $\phi_{12}$ are almost constant. In contrast, in the energy region of $2.0 < E_{cm} < 2.6$ MeV, both $E1/E2$ and $\phi_{12}$ vary rapidly as a consequence of the broad $1^-$ resonance at 9.58 MeV in $^{16}$O. As shown by Brune and Sayre (Fig. 12 of ref. [12]), in this energy region, the variation of $\phi_{12}$ leads to subtle changes in the measured angular distributions. Therefore, the region of $E_{cm} = 2.0-2.6$ MeV, is ideally suited for testing the accuracy of measured angular distributions.

However, Assunção et al.[6] observed substantial disagreement with the theoretical prediction for energies down to 1.31 MeV, including the region of interest $E_{cm} = 2.0-2.6$ MeV[2] (cos $\phi_{12}$ differs by up to a factor 2). Ouellet et al.[4] noted in Table II (footnote b) that they were unable to measure $\phi_{12}$ from 1.9−2.4 MeV. The data of Redder et al.[3] are measured mostly with 100% error bars in this region, as are the data of Dyer and Barnes[2]. Thus, so far, no available data at $E_{cm} = 2.0-2.6$ MeV exhibit the predicted[2] strong variation of $\phi_{12}$ over the $1^-$ resonance region.

The theoretical values for $\phi_{12}$ were originally considered to be a prediction of R-Matrix theory[11]. However, more recently, it was shown that the theoretical prediction for $\phi_{12}$[2] is a consequence of the Watson theorem, which is derived assuming the unitarity of the scattering matrix[13,14]. This theoretical prediction is valid in general when the capture cross-section is small, and it is the only open reaction channel, as is the case here. Hence, the observed discrepancy between the world data and theory from $2.0 < E_{cm} < 2.6$ MeV, is a disagreement with a fundamental prediction of quantum theory, and it cannot be overlooked.

Furthermore, the modern measurements of gamma-ray data of the $^{12}$C$(\alpha, \gamma)$ reaction[6] were analysed[12,13], and large uncertainties were concluded. Large backgrounds in the measured gamma-ray spectra, induced, for example, by in-beam neutrons (e.g. see Fig. 6 of ref. [6]), leads to large uncertainties in the measured angular distributions, and the extracted $E1/E2$ cross sections[13]. The large uncertainties deduced for the modern data[6] and similar gamma-ray data[15] lead to uncertainties in the $R$-Matrix analyses and extrapolation to stellar conditions[10]. We conclude that measurements of data with low backgrounds are needed for accurate extrapolation to stellar energies.

In this work, cross sections and angular distributions of the $^{12}$C$(\alpha, \gamma)^{16}$O reaction are obtained by measuring the inverse $^{16}$O$(\gamma, \alpha)^{12}$C reaction with gamma-beams and a Time Projection Chamber (TPC) detector. In sharp contrast to measurements with gamma-ray detectors, our data are measured with a vanishingly small intrinsic background. Furthermore, when measuring with a traditional array of gamma-ray detectors, an accurate knowledge of the relative efficiencies of the different detectors is crucial. Using our method, angular distributions are measured in one detector and the angular efficiencies are easily simulated. We obtain cross sections in agreement with current world data and further evidence the strength of our method with accurately measured angular distributions, leading to extracted $\phi_{12}$ values in general agreement with the trend predicted by unitarity. This serves as strong motivation to extend the initial measurements reported here to lower energies, where measurements are even more demanding.

## Results

Detailed angular distributions of the reaction were measured with an unprecedented event-by-event resolution of ~2°, and over almost a full range of polar angles, using a TPC detector with an optical readout (O-TPC). We note from the outset that our measurements agree with current world data for the total reaction cross-section. We also observe similarly shaped angular distributions, and, in contrast to some previous work, an agreement with the predicted variation of $\phi_{12}$. Our initial success in measuring angular distributions, reported here, serves as a "proof of principle" for our method and it encourages us to extend our measurements to lower energies, using a TPC detector[16] and gamma-ray beams from the HI$\gamma$S facility[17,18] in the USA and the newly constructed ELI-NP facility[19,20] of the EU.

**Inverse process (detailed balance).** In our experiment, instead of measuring the fusion of $\alpha + ^{12}$C to form $^{16}$O, we use gamma-ray beams to measure the time reversed process: the $^{16}$O$(\gamma, \alpha)^{12}$C photo-dissociation reaction. The photo-dissociation cross-section is directly related to the capture cross-section via the principle of detailed balance: $\sigma(\gamma, \alpha) = k_\alpha^2/2k_\gamma^2 \times \sigma(\alpha, \gamma)$ where $\hbar k_\alpha = \sqrt{2\mu E_{cm}}$, and $\hbar k_\gamma = \hbar\omega/c = E_\gamma/c$, and is larger by a factor of ~50. We measure the tracks of the emanating $\alpha$ and $^{12}$C in a TPC detector operating with $CO_2$ gas[21]. The $^{16}$O photo-dissociation events are unambiguously identified and measured in a ~$4\pi$ geometry with high efficiency and a vanishingly small intrinsic background, as shown in Fig. 1 and in the "Methods" section (see below). Since the angular distributions are measured in a single detector, the relative corrections between different angles (which are essential for measurements with a gamma detector array) are not necessary. These factors allow us to measure accurate angular distributions of the $^{12}$C$(\alpha, \gamma)$ reaction, as we report here.

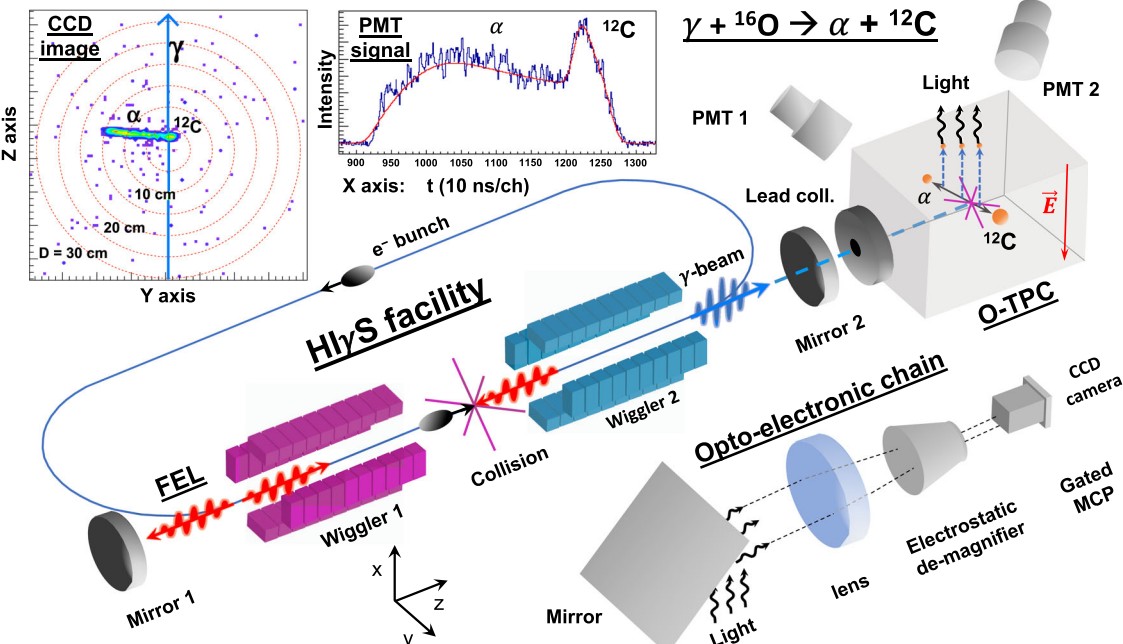

**Fig. 1 Schematic diagram of the experimental setup used at the HIγS facility at Duke University.** Not drawn to scale. The distance between the two mirrors is 53.73 m, between the FEL-electron collision point and the collimator is 60 m, and between the collimator and the O-TPC is 20 m. We show the FEL-electron ring, the Compton Backscattered gamma-ray, the O-TPC with the PMTs and the opto-electronic chain, discussed in the text. In the upper left corner we show a typical CCD image of a track and a PMT signal with the fitted lineshape of tracks from an $\alpha$ and $^{12}$C. The three-dimensional track is reconstructed with the $(y, z)$ coordinate measured by the CCD image and the $x$ coordinate calculated using the time projection signal measured by the PMT, as discussed in the text.

**Gamma beam**. Intense, quasi mono-energetic photon beams were produced at the High Intensity γamma Source (HIγS) of the Triangle Universities Nuclear Laboratory, as discussed in ref. [17]. Briefly, as depicted in Fig. 1, high-energy gamma beams were generated through the inverse-Compton scattering of an electron bunch with the free-electron-laser (FEL) photons, produced by the previous electron bunch[17]. The gamma-beam energy is determined by the FEL wavelength and the electron energy. A circularly polarised gamma beam was used in order to limit the wear of optical components. Data were taken using beam intensities of ~$10^8\gamma$/s with an energy spread of ~3% (~300 keV FWHM), at the following nominal beam energies: $E_\gamma = 9.08$, 9.38, 9.58, 9.78, 10.1, and 10.4 MeV.

The gamma beam was defined by a 15 cm long lead collimator with an 11mm-diameter aperture. It then passed successively through five thin scintillating paddles that were used for online measurements of the beam flux. The gamma beam then passed through air to reach a 5 $\mu$m kapton window, which isolated the gas in the O-TPC from atmosphere. This was followed by a strong permanent magnet, which deflected electrons produced in the window away from the gaseous target.

Downstream of the O-TPC, the flux was also measured by detecting neutrons from the $d(\gamma, n)p$ reaction using an in-beam D$_2$O target. Prior to each measurement, copper of various thicknesses was inserted to attenuate the beam by up to a factor of $10^5$. The remaining photons were incident on an HPGe or NaI(Tl) detector, to measure the photon energy and flux, respectively. The 10-inch NaI(Tl) detector provided the absolute number of photons in the attenuated beam, which was used to cross-calibrate the relative flux paddle detectors. The energy of beam was obtained using a large, high-efficiency HPGe detector. The measured spectra were unfolded using a Monte Carlo technique[22–24] to obtain the energy profile of the beam. The O-TPC detector was aligned with respect to the beam using a

gamma camera and lead absorbers placed in the front and back of the detector as discussed in refs. [22–24].

**The O-TPC detector**. The detector operation is discussed in ref. [21]. Briefly, it was operated with a gas mixture of CO$_2$ (80%) and N$_2$ (20%), at 100 Torr pressure. The active gas volume was $30 \times 30 \times 21$ cm$^3$. The 12 mm wide gamma beam entered the active volume through a 15mm opening. The N$_2$ gas was used to produce the near-UV (338 nm) light detected by the PMTs and the opto-electronic chain. The resulting $\alpha$ and $^{12}$C from the photo-dissociation of $^{16}$O propagated through the gas and ionised atoms along their tracks, with an energy loss profile that was calculated using SRIM[25], giving the lineshape shown in Fig. 1. The electrons drift upwards in the O-TPC under the influence of a ~200 V/cm electric field (2 V/cmTorr reduced electric field). The drift velocity of the electrons was measured to be 11.1 mm/$\mu$s, in agreement with calculations using Magboltz[26].

The drift electrons were then multiplied by a stronger electric field of 20 V/cmTorr, giving rise to an avalanche and producing light. The light was detected by four photomultiplier tubes (PMTs) that surrounded the top of the O-TPC, as depicted in Fig. 1. The PMT signals were digitised with a 100 MHz 12-bit flash ADC. The PMT signals measure the arrival times of the drift electrons, recording the *time projection* of the track, which allows us to measure the $x$-coordinate, as shown in Fig. 1. At the same time, optical photons propagate through the opto-electronic chain and are focussed onto a CCD camera, which photographs the track. An image of a typical track from the $^{16}$O$(\gamma, \alpha)^{12}$C reaction is shown as the left figure in Fig. 1. The photograph allows us to measure the (in-plane) $y$–$z$ coordinates of the track. The combination of the time projection $(x)$ and CCD image $(y,z)$ provides the three-dimensional coordinates of the track, from which the scattering angles were determined.

The anode signal, track length, and the total light signal were used to measure the total energy deposited, event-by-event, in the TPC. Tracks in the O-TPC were calibrated using the 3.183 MeV $\alpha$ particles from a standard $^{148}$Gd radioactive source. The energy resolution was measured to be 4% (FWHM). The particle identification was achieved by measuring $dE/dx$ of the particles along the track. These lineshapes were used to determine the photo-dissociation events, as shown in Fig. 1.

**Background.** Events recorded by the O-TPC include Compton electrons, cosmic rays, $^{14}$N$(\gamma, p)$, $^{16,18}$O$(\gamma, \alpha)$, and $^{12}$C$(\gamma, \alpha)^8$Be reactions. All background events from Compton electrons, cosmic rays, $^{14}$N$(\gamma, p)$ and $^{18}$O$(\gamma, \alpha)$ were easily removed. Compton electrons deposit up to 100 keV total energy in the O-TPC, and were removed by an 800 keV electronics threshold on the anode signal. The majority of the cosmic events were removed by inspecting the track image and requiring an interaction point which is within ± 6 mm of the centre of the gamma-beam position. Events from the $^{14}$N$(\gamma, p)$ reaction were removed by identifying the $dE/dx$ lineshape of the proton, which differs significantly from the $\alpha$ particles. The $^{18}$O$(\gamma, \alpha)$ reaction events deposit 934.95 keV more energy than the $^{16}$O$(\gamma, \alpha)$ events, and were easily removed by measuring the total energy deposited in the O-TPC.

However, the energy deposited by the $^{12}$C dissociation events is only 112.85 keV lower than for $^{16}$O dissociation events. Hence, the $^{12}$C dissociation events could not be removed, since the beam resolution is ~300 keV. However, it is worth noting that when the $^{12}$C and $^{16}$O dissociation events are identified, as seen below, we could still measure the reaction with the detector resolution (~100 keV), since the beam energy could be evaluated event-by-event, by adding the known Q-value to the energy deposited in the detector by the reaction products. We did not carry out this procedure here, due to the current low statistics.

**Lineshape analysis.** The analysis of the $^{12}$C$(\gamma, \alpha)$ events was already reported[27,28]. Namely, the measured PMT lineshape was fitted with the lineshapes predicted for $^{12}$C and $^{16}$O dissociation events, and the resulting reduced $\chi^2/\nu$ of each best fit was derived. In Fig. 2, we show a two-dimensional lego plot of the obtained reduced $\chi^2/\nu$, demonstrating a clear separation of the $^{12}$C and $^{16}$O dissociation events with vanishingly small intrinsic background. This was followed by a visual inspection of the remaining events to remove the small number of background events.

The reconstructed three-dimensional track allowed us to measure the scattering angle and azimuthal angle of each event. The CCD image measured the in-plane angle ($\alpha$), and the PMT

signal the out-of-plane angle ($\beta$), from which the scattering angle ($\theta$) and azimuthal angle ($\phi$) were calculated[21]: $\tan\phi = \tan\beta/\sin\alpha$ and $\cos\theta = \cos\beta \times \cos\alpha$.

The efficiency for detecting $^{16}$O$(\gamma, \alpha)$ events was calculated using Monte Carlo simulations. We define the fiducial volume over $|\beta| < 55°$ to exclude the region where the scattering angle carries the largest uncertainty. Over the selected fiducial volume, we obtained an event-by-event angular resolution better than 2°. The fiducial volume was further restricted to $|\beta| > 20°$ in order to provide the cleanest separation of $^{16}$O and $^{12}$C dissociation events. We note that the fiducial cuts lead to a vanishing (event-by-event) efficiency at the extremes of our angular distributions. This reduced efficiency is the small price paid for achieving cleanest data, in a region of angles where the cross-section is anyhow predicted to be small. The Monte Carlo simulation also accounted for edge effects, where one of the particles escapes the detector, meaning that only a section of the track is contained inside the fiducial volume. The efficiency rises from 0% to 60% for scattering angles, $\theta$, between 20° and 55°. The efficiency varies from 60% to 40% between 55° and 90°. Uncertainties in the fiducial volume cuts, including the range straggle of the $\alpha$ and $^{12}$C, give rise to a <1% uncertainty in the overall efficiency correction. The event-by-event efficiency curves, and their small systematic uncertainties, are included in the electronic supplement.

**Measured Cross Sections.** For each nominal beam energy $E_\gamma =$ 9.38, 9.58, and 9.78 MeV, we accumulated ~500 $^{16}$O photo-dissociation events. For $E_\gamma =$ 9.08, 10.1, and 10.4 MeV, we accumulated ~100 events. The actual gamma-beam energies were measured using the attenuated beam and HPGe detector. The measured beam energies, with 30 keV uncertainties, were: $E_\gamma$(HPGe) = 9.01, 9.41, 9.61, 9.78, 10.10, and 10.43 MeV. The average of the previous world data (plotted in Fig. 3) was used to calculate the "effective energies"[12], defined as the beam energy averaged over the FWHM of the broad gamma-beam, weighted by the global cross-section data. These are: $E_\gamma^{eff} =$ 9.18, 9.45, 9.63, 9.80, 9.98, and 10.44 MeV corresponding to effective centre-of-mass energies $E_{cm}^{eff} =$ 2.02, 2.29, 2.47, 2.64, 2.82, and 3.28 MeV.

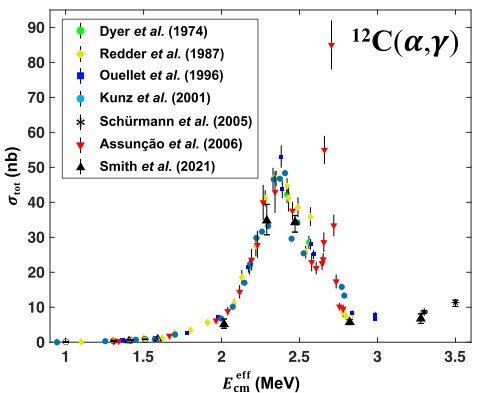

**Fig. 3 The total reaction cross sections for the $^{12}$C$(\alpha, \gamma)^{16}$O reaction.** We present the cross sections measured in this work, with a gamma-beam resolution of FWHM ~300 keV, compared with previous measurements using particle beams, with considerably better energy resolutions. The average of the shown previous world data was used to calculate the "effective energies" of our measurements with a broad gamma-beam. The error bars include 1$\sigma$ SD statistical uncertainties due to the number of events measured at each energy, and uncertainties due to the beam energy spread.

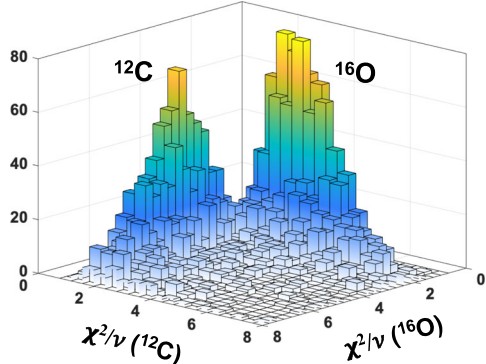

**Fig. 2 Two-dimensional Lego plot for separation of $^{16}$O and $^{12}$C events.** Here we plot the smallest deduced $\chi^2/\nu$ of fitting the PMT lineshape with the predicted lineshapes of $^{16}$O and $^{12}$C dissociation events.

The integrated total beam intensity was measured using plastic paddle detectors, which were calibrated to absolute flux using the attenuated beam and 10" NaI(Tl) detector. There is an estimated 11% uncertainty in the beam flux, due to uncertainties in the copper attenuators. The counts at each beam energy, along with the beam flux and the efficiency, were used to calculate the total cross-section of the $^{16}O(\gamma, \alpha)$ reaction, from which we derived the cross-section of the $^{12}C(\alpha, \gamma)$ reaction, as shown in Fig. 3, using detailed balance. Using the method described in ref. [12], a correction factor was applied to the cross sections to account for the energy spread of the gamma beam. Systematic errors in the cross-section due to the beam energy uncertainty were estimated by examining the variation of the world data cross-section ±30 keV about each $E_{cm}^{eff}$. In Fig. 3, the error bars contain statistical uncertainties (4−15%) and uncertainties due to the beam energy, but exclude the global 11% uncertainty on the beam intensity.

An overall agreement with the previously measured total cross-section allows us to benchmark our results against previous measurements. It is essential to benchmark our data at energies above 2.0 MeV where the total cross-section (over the $1^-$ resonance) is large and determined with high accuracy, with an agreement among measurements, as we show in Fig. 3. This benchmarking is *required* at this stage to further demonstrate the validity and strength of our method. It further motivates extending our measurements to lower energies, which is made possible by major investments in developing new initiatives[18] and new facilities[19,20].

**Fitted angular distributions**. The scattering angle in the laboratory frame was converted to the centre-of-mass scattering angle (shift < 2°) and saved for each event. Data were then fitted with the partial wave decomposition[2]

$$W(\theta) = (3|A_{E1}|^2 + 5|A_{E2}|^2)P_0(\cos \theta)$$
$$+ \left(\frac{25}{7}|A_{E2}|^2 - 3|A_{E1}|^2\right)P_2(\cos \theta)$$
$$- \frac{60}{7}|A_{E2}|^2 P_4(\cos \theta)$$
$$+ 6\sqrt{3}|A_{E1}||A_{E2}|\cos \phi_{12}\left[P_1(\cos \theta) - P_3(\cos \theta)\right],$$

by varying all three fit parameters: $A_{E1}$, $A_{E2}$, and $\phi_{12}$. The theoretical angular distributions were convolved with a Gaussian to account for a 2° angular uncertainty. All fits utilised the unbinned maximum likelihood method in order to negate the effects of angle binning and preserve our excellent angular resolution. The solid fit lines in Fig. 4 show the optimised partial wave decompositions convolved with the angular resolution, and include angular distribution attenuation factors ($Q_i$) to account for angle binning[2]. For 20° bins, the obtained $\chi^2/\nu$ fit values are 0.17, 1.62, 1.51, 4.79, 5.89, and 0.85 at $E_{cm}^{eff}$ = 2.02, 2.29, 2.47, 2.64, 2.82, and 3.28 MeV, respectively. We note that the $\chi^2$ values are quite sensitive to the choice of angle binning. Due to the 2.64 MeV point's proximity to the very narrow $2^+$ resonance at 2.68 MeV, this angular distribution was fit as a sum of a pure E2 angular distribution (40%) and the partial wave decomposition (60%)[2]. The relative contribution of the narrow E2 resonance was determined by examining the total energy deposited in the detector for all identified $^{16}O$ events at this beam energy, which was fit with a sum of two Gaussians.

## Discussion

The obtained $E1-E2$ mixing phase angles ($\phi_{12}$) are shown in Fig. 5. The theoretically predicted $\phi_{12}$[2], discussed above, averaged

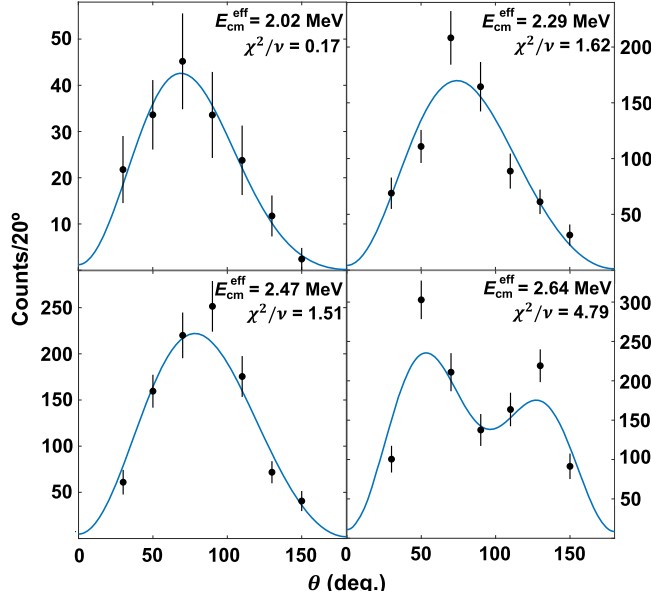

**Fig. 4 Measured angular distributions of the $^{12}C(\alpha, \gamma)^{16}O$ reaction.** Plotted data include efficiency corrections and are presented at the shown "effective" centre-of-mass energies, with the three parameter fit ($|A_{E1}|$, $|A_{E2}|$, and $\phi_{12}$) of the partial wave decomposition. The $1\sigma$ SD error bars show statistical uncertainties due to the number of events measured in each angle bin.

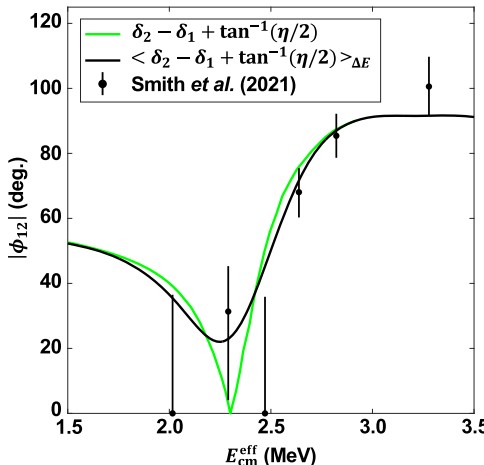

**Fig. 5 The measured E1-E2 mixing phase angles.** The measurements of $\phi_{12}$ are compared to the phase angles predicted by unitarity[13, 14] (green line), and the prediction convolved with the gamma-beam energy resolution of FWHM ~300 keV (black line). The error bars show $1\sigma$ SD statistical uncertainties extracted from the fits to the angular distributions.

over the beam energy resolution (FWHM ~300 keV), is also plotted in Fig. 5. Our measured $\phi_{12}$ values are in general agreement with the trend predicted by unitarity, exhibiting a strong variation across the $1^-$ resonance region.

In conclusion, we have presented a measurement of the $^{12}C(\alpha, \gamma)^{16}O$ reaction, using an entirely different approach to previous experimental efforts, with different systematic errors. Our method permitted us to measure angular distributions spanning the energy region $E_{cm}$ = 2.0−2.6 MeV, with improved precision, as for example in comparison with Assunção et al.[6]. This demonstrates the power of our experimental approach as a

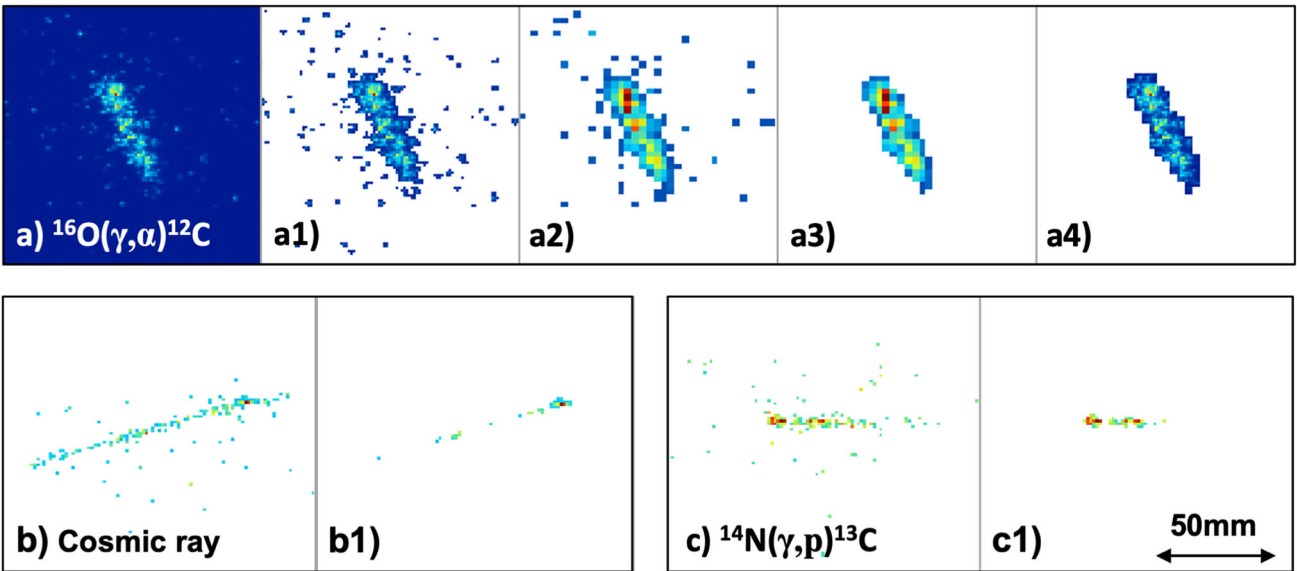

**Fig. 6 Image processing steps for the photo-dissociation and cosmic events. a** $^{16}O(\gamma, \alpha)^{12}C$, **b** cosmic ray, **c** $^{14}N(\gamma, p)^{13}C$.

promising tool to investigate the cross-section of the $^{12}C(\alpha, \gamma)^{16}O$ reaction at gamma-beam facilities.

## Methods

The O-TPC allows us to reconstruct, in three dimensions, the events corresponding to the $^{16}O(\gamma, \alpha)^{12}C$ reaction, by measuring all information pertaining to the reaction[21]. It measures the total energy deposited, the particle type, momentum, and angular distributions of the emitted particles. Thus, it allowed us to differentiate different reactions using three main tools: the photograph taken by the CCD camera, the total energy deposited by each event, and the time projection of each track. Further details of the analysis methods are provided below.

**Image processing**. At the beginning of each run, a photograph of the TPC was taken without the incident beam, which was subtracted from each image of a track. An example $^{12}C + \alpha$ track (zoomed), after this correction, is shown in Fig. 6a). The average background pixel value, $\bar{p}$, and its standard deviation, $\sigma_p$, were quantified, and a threshold of $\bar{p} + 5\sigma_p$ applied. All pixel values below this threshold were set to zero. This results in Fig. 6a1). The resolution of the image was then degraded by a factor of 4 in each direction, resulting in Fig. 6a2). The *hot* pixels surrounding the main track correspond to partial tracks of electrons that scatter during the ionisation. These are removed by zeroing the pixel values in the compressed image that do not have 5 or more non-zero neighbouring pixels. This gives Fig. 6a3). Finally, with a mask provided by the remaining non-zero pixels, the original resolution of the image was restored to give Fig. 6a4).

For highly ionising $^{12}C + \alpha$ and $^{14}C + \alpha$ tracks, corresponding to photo-dissociation of $^{16}O$ and $^{18}O$, respectively, the final image consists of a single cluster of non-zero pixels on a blank background. Sparks in the TPC generate large numbers of clusters and are readily removed. Proton tracks induced by cosmic rays and from the photo-dissociation of $^{14}N$, shown in Fig. 6b, c, respectively, result in two or more clusters of non-zero pixels. This is because the stopping power of the proton is much lower than the $^{12/14}C + \alpha$. This leads to low pixel values, closer to the background level. Only tracks with ≤2 clusters of pixels were taken for further analysis. A second data reduction cut was placed, demanding that the origin of each track lay within ±6 mm of the beam position. These two data analysis cuts removed ~70% of raw events.

From each image, the angle of the track in the $y–z$ plane, relative to the beam direction, $\alpha$, was extracted. The $y–z$ coordinates of each pixel in a track were plotted and a linear fit was performed. The error on the gradient was extracted using standard techniques, which led to a typical uncertainty on the extracted $\alpha$ angle of ~2°.

**Track length and energy deposition**. For each event, the two pixels with the largest separation were used to deduce the length of the track in the $y$-$z$ plane of the coordinate system shown in Fig. 1. The $x$ length was extracted from the time projection. These were combined to obtain the total track length. Additionally, the energy deposited by each event in the TPC was measured using the total pulse height. As expected, the track length and energy were correlated. A 2D plot of track length vs. energy is shown in Fig. 7. Regions of high intensity in this plot correspond to different reactions. With similar $Q$-values, the $^{12}C$ and $^{16}O$ photo-

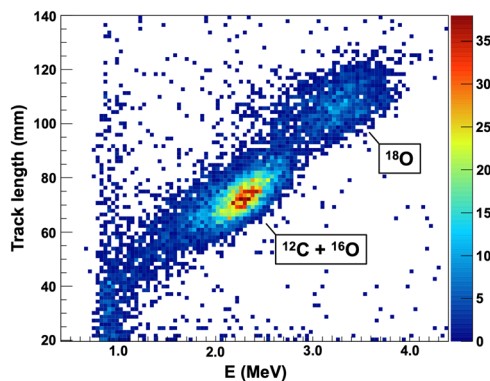

**Fig. 7 Histogram of track length vs. deposited energy.** A clear separation of $^{16}O/^{12}C$ and $^{18}O$ can be made.

dissociation events appear within the same peak. The $^{18}O$ photo-dissociation $Q$-value differs sufficiently (935 keV) that these events appear as a separate peak. A 2D software cut was placed around the $^{12}C/^{16}O$ peak to remove all other types of events.

**Lineshape analysis**. In order to differentiate between the remaining $^{12}C/^{16}O$ events, and to extract the out-of-plane angle, $\beta$, a lineshape analysis was performed. The shape of the time projection depends on the stopping powers of the reaction products along with the $\beta$ angle of the track. Tracks with large $\beta$ result in long time projections, and those with small $\beta$ give short time projections, corresponding to the track lengths in the $x$-direction. The SRIM software[25] was used to determine lineshapes for each type of photo-dissociation event: $^{16}O \rightarrow {}^{12}C + \alpha$ and $^{12}C \rightarrow 3\alpha$. These were then projected for $\beta$ angles ranging between ±90°. Each of the 180 $\beta$ projections for both $^{12}C$ and $^{16}O$ were fitted to the time projection of each event. The $\beta$ angle of the event was obtained through a $\chi^2$ minimisation. Differentiating between $^{12}C$ and $^{16}O$ events was achieved by examining the relative $\chi^2/\nu$ of the $^{16}O \rightarrow {}^{12}C + \alpha$ and $^{12}C \rightarrow 3\alpha$ best fits, as shown in Fig. 8. A histogram of the relative $\chi^2/\nu$ values was shown earlier in Fig. 2. A diagonal software cut through the centre of Fig. 2 removed the majority of $^{12}C$ events from the data, preserving, on average, 95% of $^{16}O$ events.

**Visual inspection**. All remaining events underwent a visual inspection, to remove the small number of $^{12}C$ events that escaped earlier software cuts. Along with the time projection, the longitudinal projection of the image of each track was also fitted with the $^{16}O \rightarrow {}^{12}C + \alpha$ lineshape projected into the $y–z$ plane, which provided a further dimension for event rejection.

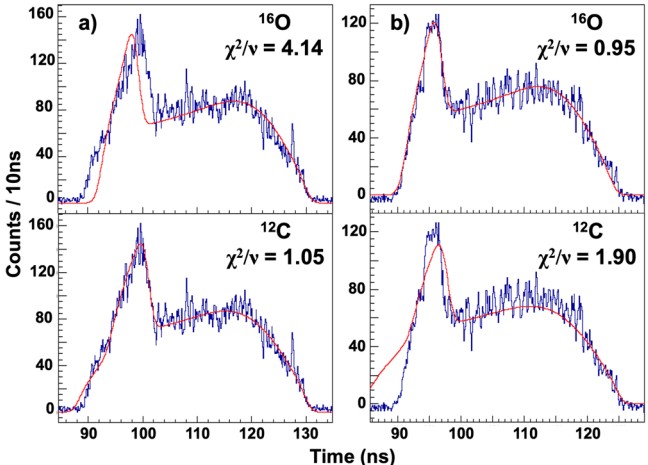

**Fig. 8 Example lineshape fits used to distinguish $^{12}$C/$^{16}$O events. a** A typical $^{12}$C photo-dissociation time projection. **b** A typical $^{16}$O photo-dissociation time projection. The upper and lower plots in each panel show the best fits of $^{16}$O and $^{12}$C photo-dissociation theoretical lineshapes, respectively.

## Data availability

The total cross-section data, angular distribution data, angular efficiency corrections, parameters extracted from each of the fits, and a table of systematic uncertainties, are provided as an electronic supplement to this paper. For $E_{cm}^{eff}$ = 2.02, 2.29, 2.47, 2.64, 2.82, and 3.28 MeV, lists of the measured $\theta$ angles are provided to permit an unbinned maximum likelihood analysis. The raw data have been made publicly available for re-analysis under a Creative Commons BY-NC 4.0 licence. The image files of tracks on an event-by-event basis, the ROOT files containing the time projections and energy signals, and the ROOT files containing the beam intensity monitoring data are provided. All data may be found on the Sheffield Hallam University data repository at: https://doi.org/10.17032/shu-180022.

## Code availability

Example analysis codes, written in root C++, are available in this data repository: https://doi.org/10.17032/shu-180022.

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

## Acknowledgements

The authors would like to acknowledge the efforts of the UConn-TUNL collaboration for the collection and dissemination of data. We thank the staff of HI$\gamma$S at Triangle Universities Nuclear Laboratory for the operation of the facility, and C. R. Howell for in-depth discussions and assistance with various aspects of the work. Finally, we thank the reviewers for their feedback on our research. In particular, reviewer 1 raised a significant number of specific points that have improved the manuscript. This material is based on work supported by the U.S. Department of Energy, Office of Science, Office of Nuclear Physics grants DE-FG02-94ER40870, DE-SC0005367, DE-FG02-97ER41033, and DE-FG02-91ER40608.

## Author contributions

M.G. served as the spokesperson of the UConn-TUNL collaboration[21,27]. M.W.A. assisted in data collection. The data were subsequently analysed by R.S. with the assistance of S.R.S. and D.K.S. All authors contributed to the final manuscript.

## Competing interests

The authors declare no competing interests.
