## [Peer Review File · Nature Communications]

Precision measurements on oxygen formation in stellar helium burning with gamma-ray beams and a Time Projection ChamberREVIEWER COMMENTS

Reviewer #1 (Remarks to the Author):

This paper reports on measurements of the $^{16}\text{O}(\gamma, \alpha)^{12}\text{C}$ using a novel technique.

This reaction is an extremely important one for understanding the evolution of massive stars, the formation of black holes, and the abundance of oxygen and other elements in the universe. Despite considerable effort, there is considerable uncertainty in the reaction rate.

The cross section results reported in this paper are similar to ones reported previously that are referenced in the paper. In this sense, the results are not particularly impactful. It is likely that the angular distributions reported here have reduced systematic errors, compared to previous measurements. More importantly, this paper reports the first results using a very new and different technique. Considering the likely improvements in the detector, beam intensity, beam time, data analysis, etc..., it is very probable that this technique will in the future yield results that surpass the quality of the currently available data. In this sense, this paper is an important development and represents a significant milestone.

In my view, this latter point makes these results noteworthy and valuable to publish. I have some specific criticisms of the paper which are listed below. All of these are correctable via revision. The authors should keep in mind other scientists will want to incorporate this data in their global fits. It is thus essential that the systematic

uncertainties are carefully documented and that the results are made available in a form and with sufficient documentation that they can be fitted by others.

1. The last sentence in the abstract, “These allowed us to measure, for the first time, the interference angle of the $\ell = 1$ and 2 partial waves contributing to this reaction (φ_{12}), which agrees with predictions based on the unitarity of the scattering matrix.” is very misleading. This quantity has been measured several times before, in Refs. [2-4,6] of this paper. Although some of the previous measurements disagree with unitarity, that is not the case for all of the previous measurements. Nor is it necessarily the case that the present measurements have the smallest uncertainties. I suggest the authors modify the statement in some way to make it factual or remove it.

2. Line 42. This paper quotes Ref. [8] as an example of a modern analysis that quotes a 100% uncertainty for the reaction rate. However, Ref. [8] should not be in this discussion at all. The focus of that paper was what $^{16}\text{O}(e,e')$ can bring to the table and not on evaluating a best value and uncertainty for the low-energy S factor. Although the paper does show a figure with a very large error bar for the $^{12}\text{C}(\alpha,\gamma)$ S-factor at low energy, it is not discussed at all and no information about statistical procedures, R-matrix parameters, or comparison to other work is given. An excellent and comprehensive review is given by Ref. [9] of the present work. Other useful sources are An et al., Phys. Rev. C 92, 045802 (2015), Xu et al. (NACRE II), Nucl. Phys. A918, 61 (2013), and other works mentioned

in Sec. VIII of Ref. [9].

3. Line 72. The paper states “For example, it was concluded that the centre of mass motion of the alpha + 12C system leads to large shifts of the centre of mass angles, which were previously ignored [12].” And the paper states starting on line 114 regarding the reverse reaction “This process does not require significant centre of mass corrections.” The authors appear to have some misconceptions here. The point in Ref. [12] is not that center of mass motion is a large effect overall, but that it has significant effects on fitting the angular distribution. This occurs because it effects the data systematically with angle, mimicking the forward-backward asymetry that E1-E2 interference produces. I see no reason to believe these considerations are any less important in the reverse reaction. A quick calculation shows that they are comparable in magnitude:

$\theta_{(\gamma, \text{lab})} \theta_{(\gamma, \text{cm})} \frac{d\Omega_{(\text{cm})}}{d\Omega_{(\text{lab})}} \theta_{(\alpha, \text{lab})} \theta_{(\alpha, \text{cm})} \frac{d\Omega_{(\text{cm})}}{d\Omega_{(\text{lab})}}$

5.00 5.05 1.02 5.00 5.01 1.04

90.00 90.62 1.00 90.00 91.16 1.00

This suggests this effect is likely non-trivial in the reverse reaction. One would need to also look at the 12C transformation and take into account that the α and 12C are not exactly back-to-back in the lab system.

4. Line 136. The paper provides conflicting information about how the beam energy is determined. It is first said that “The gamma beam energy is determined by the FEL wavelength and the electron energy.” Later, line 161, it is said “The energy of beam was obtained using a large, high-efficiency HPGe detector.” This situation should be clarified. Are the wavelength and electron energy not known with sufficient accuracy?

5. Line 229. The paper states “Hence, the ^{12}C dissociation events could not be removed by relying solely on the total energy deposited in the O-TPC, which was measured with ~ 100 keV resolution.” Perhaps I am missing something here, but if the photon beam resolution is 3% or ~ 300 keV, aren’t the peaks spread out by that factor, making a separation based upon energy hopeless? I feel that paper needs to be more clear on the various resolutions here and what the effective resolution actually is.

6. Line 271. The uncertainty in the photon beam energy is quoted to be 30 keV. The authors should point out that this is a non-negligible uncertainty where there cross section changes quickly with energy. For example, at $E_{\text{(c.m.)}} = 2.29$ MeV, it appears to correspond to about a 15% change in the cross section.

7. Line 274. The term “effective energy” is nowhere defined in the paper. Perhaps it refers to one of the schemes described in Ref. [12]?

8. Lines 282-290. The paper needs to be more explicit about the systematic error budget. The 11% uncertainty in beam flux is clear.

The paper then refers to a “global 11% uncertainty.” Are all remaining systematic uncertainties ($^{12}\text{C}(\gamma, \alpha)$, other backgrounds, the efficiency correction,...) negligible compared to 11%? These data will likely be utilized by others in the future. Poorly documented uncertainties has historically been a major impediment to this.

9. Fig. 3. There are two data sets that are not included in this figure that I feel should be. They are slightly lower in energy than the present work, but they represent some of the most recent and highest quality angular distribution data. The references are Makii et al., Phys. Rev. C 80, 065802 (2009), and Plag et al., Phys. Rev. C 86, 015805 (2012).

10. Fig. 3. The data source “Schürmann et al. (2005)” should be 2011, i.e., Ref. [7]. And the error bar on the 3.3-MeV point from this source seems to be garbled.

11. Fig. 3, caption. It is stated that the gamma beam resolution is ~ 120 keV. Elsewhere in the paper, 300 keV is given for this quantity.

12. Fig. 4. I am confused by the presentation of the data and fit. The text states that the data were put into 20° bins and then fitted by convolving the theory with a 2° Gaussian for the angular uncertainty and the 20° bin width. But the theory curve shown vanishes at 0° and 180° , which can only be the case if no angular convolution is present. Have the data shown somehow been deconvoluted for the binning and angular resolution? Further

explanation is needed.

13. Line 308. The authors state “The low statistics (a factor of ~ 5 smaller) of the angular distribution measured at 2.02 MeV lead to a non statistical ensemble of data points, and the resulting fit is over-determined. However, we still use the usual statistical procedure (minimum $\chi^2 + 1$) to quote the uncertainties of the extracted fit parameters.” This statement suggests the authors are not using the appropriate statistical methods and that perhaps the lowest energy should be disregarded. I believe this fitting problem is ideally suited for the Unbinned Maximum Likelihood method. In any case, this part of the analysis is weak and should be improved. Later in the paper, around line 324, the authors discuss the lowest-energy point, which lies several standard deviations from their unitarity prediction. The authors attribute this to the low statistics of the measurement. But again, this really seems to be pointing to incorrect statistical procedures being applied.

14. I urge the authors to supply as an electronic supplement to the paper including the cross section results from the present experiment in Fig. 3 and the data underlying Fig. 4. The files should include enough information so that somebody else could reproduce the results given in this paper or include these data in their own R-matrix fit. The availability of the raw data mentioned in lines 441-452 is also useful, but for most users, having the processed data available in a form where they can include it in a global analysis would be the most helpful.

15. Line 317. The paper refers to “The theoretically predicted ϕ_{12} ,

discussed above..." but I do not see a reference or calculation of this quantity anywhere in this paper.

16. Line 329. The paper states "Most importantly, we observe for the first time the predicted dip at 2.4 MeV, unlike previous measurements [6]." There are several issues with this statement. The first is that the observation of a dip in the present experiment depends upon critically on the lowest-energy data point, which has been extracted using questionable statistical methods. Also, the experiment of Redder et al. [3] observed the dip previously.

17. Line 333. The paper states "Our method permitted us to measure angular distributions with high precision and, for the first time, we deduce the E1-E2 mixing phase (φ_{12}), which is in agreement with the prediction of fundamental theory based on unitarity of the scattering matrix." See comments above regarding a similar statement in the abstract.

Reviewer #2 (Remarks to the Author):

The paper is very well written, has nice figures, presents results of a very challenging data analysis, and makes some good arguments towards the importance of the $^{12}\text{C}(\alpha,\gamma)$ reaction rate, but I have a couple of issues with it.

1) I think the authors emphasizes the wrong points. Their "big claim" is that they finally measure a value of the E1/E2 mixing angle that is consistent with that obtained from elastic scattering phase shifts for the first time. First, I don't think this is true. I think the recent measurement by Plag et al. (2012) and Makii et al. (2009) also probably do this, they just don't emphasize it. I believe the important contribution of this paper is that the group is pioneering a completely different way of measuring the

$^{12}\text{C}(\alpha,\gamma)$ reaction with completely different systematic uncertainties. This publication shows that they have benchmarked it well against previous measurements and now they are ready to go to lower energies.

2) Is a first measurement of $^{16}\text{O}(\gamma,\alpha)$, which is in good agreement with previous results, enough for Nature? I'm not sure. It is a very nice measurement and they go into a lot of details and seem pretty upfront about any drawbacks of the analysis, but they fail to demonstrate any impact beyond what previous authors have shown. They agree with existing data, the extrapolation into the stellar temperature range remains unchanged, the impact on stellar burning, white dwarf C/O ratios and supernova ignition will remain the same, I think the paper be more relevant for an experimental oriented journal such as PRC, at best PRL.

3) The authors don't show a comparison with existing data to evaluate the improvement in quality. The data shown in figure 4 should be compared to previous attempts to measure the angular distribution. Within the energy range covered by this experiment there have been previous angular distribution measurements that they can compare with and add it to Fig. 4. Also Fig. 5 lacks previous data for comparison?

Some more detailed points with respect to the Nature checklist.

1) The noteworthiness of the results is that this paper presents the first $^{16}\text{O}(\gamma,\alpha)$ measurement of high quality. However, the authors failed to demonstrate a real improvement to previous $^{12}\text{C}(\alpha,\gamma)$ measurements. The study also is constraint to higher energies and does not extend into the critical lower energy range.

2) I think this work may lead to future measurements that will be very significant to the field, but the data presented here are not.

3) Again, more comparison with previous data is needed to back up the here presented claims.

4) The analysis has been very challenging and is excellent.

5) I think the methodology is excellent.

6) The authors give a lot of detail in this paper. It is quite impressive work, but it is not at a level to modify or even question existing claims on the low energy cross section and its impact on astrophysical environments.

Reviewer #3 (Remarks to the Author):

The carbon/oxygen ratio at the near end of explosive stellar evolution processes has major consequences for the fate of supernovae remnants. The reaction $^{12}\text{C}(\alpha,\gamma)^{16}\text{O}$ is crucial to determine this ratio. Because nuclear physics experiments are difficult, expensive, and take many years of preparation, the progress made to reach the low ^{12}C - α relative motion energies has been daunting. Disentangling the E1 and E2 photon multipolarity component in this reaction has also been plagued by low statistics, near impossibility to reach the required low energies, and by uncertainties in the parameters used in theoretical models (e.g., R-matrix theory), which serve as guide for the experimental data to low energy extrapolation.

With the advent of newer and more powerful experimental facilities using real photons (e.g., the ELI facility in Romania), the possibility to explore the inverse reaction, namely, the $\gamma + ^{16}\text{O} \rightarrow \alpha + ^{12}\text{C}$, became feasible. The challenge then becomes the pursue of an accurate detection of the four-momenta (or energies + angular distributions) of final products and the reproduction of their relative motion energy.

This manuscript reports the very first experiment performed in the ELI facility for this purpose. It focus on a novel method to determine the interference angle of the $l = 1$ and 2 partial waves contributing to the reaction. A major accomplishment of this experiment is to clarify a long standing puzzle related to the deviation that some past experiments (based on alternative methods) observed when confronted to robust theoretical predictions based on the unitarity of the scattering matrix (as reported, e.g., in Refs [10,11]).

The paper is very well written and presents a detailed description of the experimental methods used. Frankly, I cannot tell if there are any flaws in the analysis involving detection efficiency, cross-talks, correlations, background suppression, etc.

This work has immense consequences for the small steps to understand this reaction, with (a) a novel experimental approach, (b) a brand new experimental facility, (c) the resilience and competence of the authors in pursuing this goal for many years (decades) and (d) the new results on the mixing angle, clarifying the validity of known predictions. Because of that, I have no doubt that this work merits publication in this journal. I fully recommend it for publication. However, it is of my opinion that small corrections/addenda would be in place before publication.

The manuscript is well structured and easy to understand by non-experts (a subjective concept). Here are some clarifications that might be necessary.

1- Line 72: "For example, it was concluded that the centre of mass motion of the

alpha + ^{12}C system leads to large shifts of the centre of mass angles, which were previously ignored [12]."

I haven't checked Ref. [12]. But isn't this statement a bit confusing? Is there an obvious reason for that (maybe found in [12])? The authors could add a sentence to explain the reason, if that is easy.

2- The authors mention repeatedly the role of the detailed balance theorem, pointing to Eq. (9) of Ref. [17]. Isn't it much better to cite a textbook here? There are many available, some as old as nuclear physics itself.

3- Line 322: "... and exhibit the correct trend below this energy."

Maybe a reference for "correct trend" (Ref. [?]). Or it might be replaced by "predicted trend"?

4- Line 323: "... for the lowest energy angular distribution, we obtained 83 raw counts, not enough to obtain a statistically significant result."

Then why there seems to be a smaller error bar at 2.02 MeV than at other energies, for example at 3.28 MeV? Or are they unrelated?

5- Line 312: "... we still use the usual statistical procedure . . . to quote the uncertainties of the extracted parameters."

Where in the manuscript are these parameters quoted? Sorry, I could not find them.

6- Figure 5 is the main result of this experiment. Here "unitarity" might again deserve a reference.

7- I emphasize the complex filtering work has the authors carried out to achieve the final result presented in Fig. 5. But the solid curve also deserves an error band, because as mentioned in line 52, δ_1 and δ_2 are measured. Or is the final effect of their errors too small to show on ϕ_{12} ?

Reviewed by Carlos Bertulani (carlos.bertulani@tamuc.edu)

We insert below our replies to the reviewers in blue colour. Additions and changes to the manuscript are also highlighted in blue.

Reviewer #1 (Remarks to the Authors):

This paper reports on measurements of the $^{16}\text{O}(\gamma,\alpha)^{12}\text{C}$ using a novel technique. This reaction is an extremely important one for understanding the evolution of massive stars, the formation of black holes, and the abundance of oxygen and other elements in the universe. Despite considerable effort, there is considerable uncertainty in the reaction rate.

The cross section results reported in this paper are similar to ones reported previously that are referenced in the paper. In this sense, the results are not particularly impactful. It is likely that the angular distributions reported here have reduced systematic errors, compared to previous measurements. More importantly, this paper reports the first results using a very new and different technique. Considering the likely improvements in the detector, beam intensity, beam time, data analysis, etc..., it is very probable that this technique will in the future yield results that surpass the quality of the currently available data. In this sense, this paper is an important development and represents a significant milestone. In my view, this latter point makes these results noteworthy and valuable to publish.

We thank the reviewer, who together with the other reviewer stated that our “results [are] noteworthy and valuable to publish”.

We agree with the reviewer’s comments, and we thank them for the thorough review that has improved our paper. We reply to the reviewer’s “specific criticisms”, in the spirit of cooperation to improve our paper.

We note from the outset that the validity of “reports [of] the first results using a very new and different technique” must be demonstrated. As such we show in Fig. 3, the agreement of our measured cross sections with the world data. The “no-news” of Fig. 3, is in fact “excellent-news” that “impact” the very validity of our method. The benchmarking of our data with world data serves as a “proof of principle” of our new method. Furthermore, we have added to the manuscript, starting on line 344, that it is important to compare our cross section to the world data, on the 1- resonance at $E_{\text{cm}} \sim 2.4$ MeV, where the cross sections are large and the world data are consistent with each other.

On the other hand, we hope that the reviewer agrees that resolving the decades old problem of the disagreement of world data measured at 2.0 – 2.6 MeV, with the prediction of unitarity, is a “news” worthy result. We note that the relationship: $\phi_{12} = \delta_1 - \delta_2 + \tan^{-1} \eta/2$ was traditionally regarded as a property of R-Matrix theory. However, Brune [13] and Gai [14], demonstrated that this relationship is true in general since it is based on unitarity of the scattering matrix. Hence, as stated by the other reviewer, the disagreement of data measured over 2.0 – 2.6 MeV, is a disagreement with a robust prediction of quantum mechanics itself. Aside from resolving the decades old disagreement, we note that if the opposite occurred, and we confirmed the discrepancy with unitarity, it would have required alterations of current theory to analyse the data (e.g. adding more partial waves).

In consideration of the comments of the reviewer we now further clarify that we concentrate on the important energy region of 2.0 – 2.6 MeV, see below, where large variations of ϕ_{12} are predicted by unitarity. The predicted large variations of ϕ_{12} over this energy region are observed by us for the first time, and we hope the reviewer agrees that confirming the prediction of a fundamental theory is of value.

I have some specific criticisms of the paper which are listed below. All of these are correctable via revision. The authors should keep in mind other scientists will want to incorporate this data in their global fits. It is thus essential that the systematic uncertainties are carefully documented and that the results are made available in a form and with sufficient documentation that they can be fitted by others.

We note from the outset that we are making our data together with all important information available to others, with sufficient documentation as we discuss on line 505. In addition to the raw data, angular distribution and cross section data, along with a table of systematic uncertainties have been added as an electronic supplement to the manuscript.

1. The last sentence in the abstract, “These allowed us to measure, for the first time, the interference angle of the $l = 1$ and 2 partial waves contributing to this reaction (ϕ_{12}), which agrees with predictions based on the unitarity of the scattering matrix.” is very misleading. This quantity has been measured several times before, in Refs. [2-4,6] of this paper. Although some of the previous measurements disagree with unitarity, that is not the case for all of the previous measurements. Nor is it necessarily the case that the present measurements have the smallest uncertainties. I suggest the authors modify the statement in some way to make it factual or remove it.

We apologise for our less than accurate statement. We changed the abstract to read:

“We agree with current world data for the total reaction cross section and further evidence the strength of our method with angular distributions measured from $E_{cm} = 2.0 - 2.6$ MeV, where the interference angle of the $\ell = 1$ and 2 partial waves (ϕ_{12}) varies rapidly. We measure, for the first time over this energy range, ϕ_{12} values that agree with fundamental predictions based on the unitarity of the scattering matrix and reconcile the (historical) disagreement.”

We focus our attention on the energy region of $E_{cm} = 2.0 - 2.6$ MeV, where the predicted ϕ_{12} varies rapidly, since the elastic scattering phase shifts vary rapidly over the 1^- resonance. As we state starting in line 67:

“Therefore, the region of $E_{cm} = 2.0 - 2.6$ MeV, is suitable for testing the accuracy of measured angular distributions where these rapid changes lead to subtle variations in the shape of the angular distribution.”

As for the measurements of [2-4,6], we elaborate starting on line 71:

“However, in this energy region, Assuncao et al. [6] observed substantial disagreement with the theoretical prediction [2] ($\cos \phi_{12}$ differs by up to a factor 2). Ouellet et al. [4] noted in Table II (footnote b) that they were unable to measure ϕ_{12} from 1.9 – 2.4 MeV. The data of Redder et al. [3] are measured mostly with 100% error bars in this region, as are the data of Dyer et al. [2]. Thus, so far, no available data at $E_{cm} = 2.0\text{--}2.6$ MeV exhibit the predicted [2] strong variation of ϕ_{12} over the 1^- resonance region.”

2. Line 42. This paper quotes Ref. [8] as an example of a modern analysis that quotes a 100% uncertainty for the reaction rate. However, Ref. [8] should not be in this discussion at all. The focus of that paper was what $^{16}\text{O}(e,e')$ can bring to the table and not on evaluating a best value and uncertainty for the low-energy S factor. Although the paper does show a figure with a very large error bar for the $^{12}\text{C}(\alpha,\gamma)$ S-factor at low energy, it is not discussed at all and no information about statistical procedures, R-matrix parameters, or comparison to other work is given. An excellent and comprehensive review is given by Ref. [9] of the present work. Other useful sources are An et al., Phys. Rev. C 92, 045802 (2015), Xu et al. (NACRE II), Nucl. Phys. A918, 61 (2013), and other works mentioned in Sec. VIII of Ref. [9].

We agree with the reviewer and have removed the discussion of the MIT group. We instead now discuss the results of the Stuttgart group starting on line 46:

“However, Hammer et al. [11] who analysed the very same data, shown in Fig. 5 of Ref. [11], found a larger uncertainty. Two separate fits giving $S_{E1}(300) = 77.0$ keVb and 4.3 keVb, resulted in χ^2 values of 9.0 and 9.6, respectively. The small difference between these χ^2 values does not permit ruling out either of these two solutions.”

3 Line 72. The paper states “For example, it was concluded that the centre of mass motion of the alpha + ^{12}C system leads to large shifts of the centre of mass angles, which were previously ignored [12].” And the paper states starting on line 114 regarding the reverse reaction “This process does not require significant centre of mass corrections.” The authors appear to have some misconceptions here. The point in Ref. [12] is not that center of mass motion is a large effect overall, but that it has significant effects on fitting the angular distribution. This occurs because it effects the data systematically with angle, mimicking the forward-backward asymmetry that E1-E2 interference produces. I see no reason to believe these considerations are any less important in the reverse reaction. A quick calculation shows that they are comparable in magnitude. This suggests this effect is likely non-trivial in the reverse reaction. One would need to also look at the ^{12}C transformation and take into account that the α and ^{12}C are not exactly back-to-back in the lab system.

We agree with the reviewer and we removed this discussion from the manuscript. We further note that we included all kinematics effects, e.g. the (recoil) effect, and we list θ_{cm} .

4. Line 136. The paper provides conflicting information about how the beam energy is determined. It is first said that “The gamma beam energy is determined by the FEL wavelength and the electron energy.” Later, line 161, it is said “The energy of beam was obtained using a large, high-efficiency HPGe detector.” This situation should be clarified. Are the wavelength and electron energy not known with sufficient accuracy?

The gamma-beam energy is determined by the electron-FEL collision with accuracy which is not better than 30 keV. Hence, we resort to measurements of attenuated gamma-beam implanted into the HPGe. This detailed discussion is quite elaborate, and the interested reader can consult Ref. 16.

5. Line 229. The paper states “Hence, the ^{12}C dissociation events could not be removed by relying solely on the total energy deposited in the O-TPC, which was measured with ~ 100 keV resolution.” Perhaps I am missing something here, but if the photon beam resolution is 3% or ~ 300 keV, aren't the peaks spread out by that factor, making a separation based upon energy hopeless? I feel that paper needs to be more clear on the various resolutions here and what the effective resolution actually is.

We again apologise for our inaccurate narrative. What we meant to say is now clarified on line 269:

“Hence, the ^{12}C dissociation events could not be removed since the beam resolution is ~ 300 keV. However, it is worth noting that when the ^{12}C and ^{16}O dissociation events are identified, see below, we could still measure the reaction with the detector resolution (~ 100 keV), since the beam energy could be evaluated event by event, by adding the known Q-value to the energy deposited in the detector by the reaction products. We did not carry out this procedure here, due to the current low statistics”.

6. Line 271. The uncertainty in the photon beam energy is quoted to be 30 keV. The authors should point out that this is a non-negligible uncertainty where the cross section changes quickly with energy. For example, at $E_{\text{(c.m.)}} = 2.29$ MeV, it appears to correspond to about a 15% change in the cross section.

We agree, this effect is included in deducing the “effective beam energy” as we discuss in line 327.

7. Line 274. The term “effective energy” is nowhere defined in the paper. Perhaps it refers to one of the schemes described in Ref. [12]?

We define the effective energy in line 327.

8. Lines 282-290. The paper needs to be more explicit about the systematic error budget. The 11% uncertainty in beam flux is clear. The paper then refers to a “global 11% uncertainty.” Are all remaining systematic uncertainties ($^{12}\text{C}(\text{gamma},\alpha)$, other backgrounds, the efficiency correction,...) negligible compared to 11%? These data will likely be utilized by others in the future. Poorly documented uncertainties has historically been a major impediment to this.

We apologise for not discussing our systematic errors, they are very small, as we list in the attached table, which is included in our electronic data supplement. We have only one significant large overall systematic error of 11% as we discussed before in line 335. Our

statistical errors (4-15%) are considerably larger than our systematic errors and are now listed in line 342. Further details are provided in the Supplementary data file.

Table I: Systematic Uncertainties

Efficiency:	Error
Range Straggle	0.6%
Beam Energy (30 keV)	0.3%
Event Identification	2.0%
Beam:	
d(g,n)	2.5%
Overall Beam Intensity	11%

9 Fig. 3. There are two data sets that are not included in this figure that I feel should be. They are slightly lower in energy than the present work, but they represent some of the most recent and highest quality angular distribution data. The references are Makii et al., Phys. Rev. C 80, 065802 (2009), and Plag et al., Phys. Rev. C 86, 015805 (2012).

We thank the reviewer for this comment. We added these data points to our Fig. 3 and discuss these results in line 57.

1. Fig. 3. The data source “Schürmann et al. (2005)” should be 2011, i.e., Ref. [7]. And the error bar on the 3.3-MeV point from this source seems to be garbled.

We thank the reviewer for this comment. We corrected the mistake in Fig. 3.

11. Fig. 3, caption. It is stated that the gamma beam resolution is ~120 keV. Elsewhere in the paper, 300 keV is given for this quantity.

The resolution stated in Fig. 3 is for sigma of ~120 keV. The resolution stated in line 383 is for FWHM (=2.35 sigma). We changed all references to FWHM and we thank the reviewer for pointing out the inconsistency.

12. Fig. 4. I am confused by the presentation of the data and fit. The text states that the data were put into 20° bins and then fitted by convolving the theory with a 2° Gaussian for the angular uncertainty and the 20° bin width. But the theory curve shown vanishes at 0° and 180°, which can only be the case if no angular convolution is present. Have the data shown somehow been deconvoluted for the binning and angular resolution? Further explanation is needed.

We have clarified this point starting in lines 302 and 366.

The theoretical angular distribution was convolved with a 2° Gaussian to account for the resolution of the TPC. Then, in order to compare the binned data with theory, the theoretical curve was binned in the same way as the data, permitting χ^2 to be calculated and summed over each bin. The theoretical line showed in Fig. 4 is the partial wave decomposition

equation, convolved with the 2° angular uncertainty. It does not go to zero at 0° and 180° , but because the convolution is a small effect, the smearing is not visible on the scale of Fig. 4 (but it can be seen in the figure inserted below, for the $E_{cm} = 2.64$ MeV data).

It is worth mentioning that in another independent analysis by a co-author (M.W. Ahmed), the angular distributions were fitted with the traditional method of modifying the theoretical angular distribution function with the Q factors to account for the binning, and he obtained exactly the same optimised values for the E1/E2 amplitudes and ϕ_{12} .

The data points are zero at 0° and 180° due to the vanishing efficiency in these extreme regions, caused by our fiducial cuts, optimised for the cleanest separation of ^{12}C and ^{16}O dissociation events.

13. Line 308. The authors state “The low statistics (a factor of ~ 5 smaller) of the angular distribution measured at 2.02 MeV lead to a non statistical ensemble of data points, and the resulting fit is over-determined. However, we still use the usual statistical procedure (minimum $\chi^2 + 1$) to quote the uncertainties of the extracted fit parameters.” This statement suggests the authors are not using the appropriate statistical methods and that perhaps the lowest energy should be disregarded. I believe this fitting problem is ideally suited for the Unbinned Maximum Likelihood method. In any case, this part of the analysis is weak and should be improved. Later in the paper, around line 324, the authors discuss the lowest-energy point, which lies several standard deviations from their unitarity prediction. The authors attribute this to the low statistics of the measurement. But again, this really seems to be pointing to incorrect statistical procedures being applied.

We completely agree with the reviewer that “this fitting problem is ideally suited for the Unbinned Maximum Likelihood method”. We thank the reviewer for this comment. We changed the corresponding data point at 2.02 MeV in Fig. 5, and we state in line 374:

“Due to the low statistics (a factor of ~ 5 smaller) of the angular distribution measured at 2.02 MeV we analysed this angular distribution using the unbinned maximum likelihood method.”

The new fitting method does not significantly change the optimised fit parameters for this energy, but the correct procedure allowed us to extract a more realistic error bar, commensurate with the lower statistics.

14. I urge the authors to supply as an electronic supplement to the paper including the cross section results from the present experiment in Fig. 3 and the data underlying Fig. 4. The files should include enough information so that somebody else could reproduce the results given in this paper or include these data in their own R-matrix fit. The availability of the raw data mentioned in lines 441-452 is also useful, but for most users, having the processed data available in a form where they can include it in a global analysis would be the most helpful.

We are in the process of uploading the required data and information and we expect this process will be completed shortly. We are making our data together with all important information available to others, with sufficient documentation as we discuss on line 505. In addition to the raw data, angular distribution, cross section data, efficiencies, and a discussion of systematic uncertainties have been added as an electronic supplement to the manuscript.

15. Line 317. The paper refers to “The theoretically predicted ϕ_{12} , discussed above...” but I do not see a reference or calculation of this quantity anywhere in this paper.

We added Ref. [2] in line 59.

16. Line 329. The paper states “Most importantly, we observe for the first time the predicted dip at 2.4 MeV, unlike previous measurements [6].” There are several issues with this statement. The first is that the observation of a dip in the present experiment depends upon critically on the lowest-energy data point, which has been extracted using questionable statistical methods. Also, the experiment of Redder et al. [3] observed the dip previously.

We removed that statement. The statement starting in line 384 is sufficient.

However, please note that the error bars of the six data points shown in Table 2 of Redder et al. at energies 2.17–2.40 MeV, that are measured with a germanium detector have 100% error bars (as such they are only an upper limit). The data with NaI detector is essentially constant $\phi_{12} = 46\text{--}60^\circ$, and does not show the dip.

17. Line 333. The paper states “Our method permitted us to measure angular distributions with high precision and, for the first time, we deduce the E1-E2 mixing phase (ϕ_{12}), which is in agreement with the prediction of fundamental theory based on unitarity of the scattering matrix.” See comments above regarding a similar statement in the abstract.

We added on line 397 the caveat “the energy region $E_{cm} = 2.0 - 2.6$ MeV... in this energy region”.

Reviewer #2 (Remarks to the Author):

The paper is very well written, has nice figures, presents results of a very challenging data analysis, and makes some good arguments towards the importance of the $^{12}\text{C}(\alpha,\gamma)$ reaction rate, but I have a couple of issues with it.

1) I think the authors emphasize the wrong points. Their "big claim" is that they finally measure a value of the E1/E2 mixing angle that is consistent with that obtained from elastic scattering phase shifts for the first time. First, I don't think this is true. I think the recent measurement by Plag et al. (2012) and Makii et al. (2009) also probably do this, they just don't emphasize it. I believe the important contribution of this paper is that the group is pioneering a completely different way of measuring the $^{12}\text{C}(\alpha,\gamma)$ reaction with completely different systematic uncertainties. This publication shows that they have benchmarked it well against previous measurements and now they are ready to go to lower energies.

We thank the second referee for making the strong positive statement: "the group is pioneering a completely different way of measuring the $^{12}\text{C}(\alpha,\gamma)$ reaction with completely different systematic uncertainties." Here the second reviewer underlines Nature editorial policy of publication of new and "completely different way of measuring", and is echoing the other two reviewers who "recommended publication" or found our results "valuable for publication".

We thank the reviewer who joined the other reviewers in judging our work as: "pioneering completely different way of measuring the $^{12}\text{C}(\alpha,\gamma)$ reaction with completely different systematic uncertainties". As such we must demonstrate the validity of our method. We show in Fig. 3, the agreement of our measured cross sections with the world data. As stated by the reviewer the "benchmarking of our data with world data" serves as a proof of principle of our new and "completely different method". Furthermore, we added starting in line 346, that it is important to compare our cross section to the world data, on the 1- resonance at $E_{cm} \sim 2.4$ MeV, where the cross sections are large and the world data are consistent with each other.

We modified our discussion to be more specific and emphasise that we are considering the region of interest at $E_{cm} = 2.0 - 2.6$ MeV. We apologise for our less than accurate statement. We changed the abstract to read:

"We agree with current world data for the total reaction cross section and further evidence the strength of our method with angular distributions measured from $E_{cm} = 2.0 - 2.6$ MeV, where the interference angle of the $\ell = 1$ and 2 partial waves (ϕ_{12}) varies rapidly. We measure, for the first time over this energy range, ϕ_{12} values that agree with fundamental predictions based on the unitarity of the scattering matrix and reconcile the (historical) disagreement."

We focus our attention on the energy region of $E_{cm} = 2.0 - 2.6$ MeV, where the predicted ϕ_{12} varies rapidly, since the elastic scattering phase shifts vary rapidly over the 1^- resonance. As we state starting in line 67:

“Therefore, the region of $E_{cm} = 2.0 - 2.6$ MeV, is suitable for testing the accuracy of measured angular distributions where these rapid changes lead to subtle variations in the shape of the angular distribution.”

We note that Makii et al. [8] and Plag et al. [9] did not measure in the region of interest that we define above 2.0 MeV, as we discuss starting in line 57 and 63.

2) Is a first measurement of $^{16}\text{O}(\gamma, \alpha)$, which is in good agreement with previous results, enough for Nature? I'm not sure. It is a very nice measurement and they go into a lot of details and seem pretty upfront about any drawbacks of the analysis, but they fail to demonstrate any impact beyond what previous authors have shown. They agree with existing data, the extrapolation into the stellar temperature range remains unchanged, the impact on stellar burning, white dwarf C/O ratios and supernova ignition will remain the same, I think the paper be more relevant for an experimental oriented journal such as PRC, at best PRL.

It is not entirely clear to us that “the extrapolation into the stellar temperature range remains unchanged”. Over the 1^- resonance energy where the cross sections and yields are large, we point to major difficulties with current data that disagree with the predictions of unitarity. We show that this must be due to systematic uncertainties (that do not exist in our data, that are measured by a single detector). These systematic uncertainties are more dominant at low energies where the cross sections and the yields are small. And as we discuss starting in line 110:

“The large uncertainties deduced for the modern data [6] and similar gamma-ray data [14] lead to uncertainties in the R-Matrix analyses and extrapolation to stellar conditions [10].”

3) The authors don't show a comparison with existing data to evaluate the improvement in quality. The data shown in figure 4 should be compared to previous attempts to measure the angular distribution. Within the energy range covered by this experiment there have been previous angular distribution measurements that they can compare with and add it to Fig. 4. Also Fig. 5 lacks previous data for comparison?

We note from the outset concerning: “The authors don't show a comparison with existing data to evaluate the improvement in quality”. We understand that the reviewer addressed Fig. 4 in this comment, but in Fig. 3, we do compare our data to eight previous results spanning 1974 – 2012.

In addition, we previously discuss the comparison to existing data and we now expand starting in line 127:

“We note from the outset that we observe an overall agreement... with the general shape of the angular distributions. However, our measured angular distributions differ slightly from current measurements in their details. These fine, detailed differences allow us to measure ϕ_{12} values that agree with unitarity”.

We note that as demonstrated in Fig. 12 of Sayre and Brune [NIMA 698, 49 (2013)] the fit required by unitarity (two parameters fit) is only slightly different. We completely agree with Sayre and Brune that in fact underlines our statement on line 132:

“These fine, detailed differences allow us to measure ϕ_{12} values that agree with unitarity”. We do not think that a figure comparing the angular distributions will make our statement in line 132 more clear.

As for comparing to previous measurements of ϕ_{12} , we invite the reviewer to consider Fig. 11 of [6] of the world data on ϕ_{12} , that we also show below. We have also added a statement to summarise the situation starting in line 71. Adding this many data points to Fig. 5 will swamp our current data. We do not think that such a figure will help to clarify the discussion starting on line 71.

E1-E2 Mixing Phase Angle (ϕ_{12})

Some more detailed points with respect to the Nature checklist.

- 1) The noteworthiness of the results is that this paper presents the first $^{16}\text{O}(\gamma, \alpha)$ measurement of high quality. However, the authors failed to demonstrate a real improvement to previous $^{12}\text{C}(\alpha, \gamma)$ measurements. The study also is constraint to higher energies and does not extend into the critical lower energy range.

Concerning: “the authors failed to demonstrate a real improvement to previous $^{12}\text{C}(\alpha, \gamma)$ measurements”, we include a telling figure above, where we compare our results shown in Fig. 5, to the world data summarised in figure 11 of Ref. [6]. We hope the reviewer agrees this comparison demonstrates “a real improvement to previous $^{12}\text{C}(\alpha, \gamma)$ measurements”. This figure of the world data compares the results from the $^{12}\text{C}(\alpha, \gamma)$ reaction with the prediction of R-Matrix theory. However, as we discuss in the manuscript, this prediction is true in general for any theory, since it is based on unitarity of the scattering matrix as demonstrated by Brune [13] and Gai [14]. This decades old discrepancy has been ignored by the practitioners in the field. However, as we state on line 86, a disagreement with unitarity is a disagreement with

quantum mechanics, and it cannot be overlooked. Our data, on the other hand, show for the first time, agreement of the data with theory over the region of interest (2.0–2.6 MeV).

Aside from resolving decades old problem, we again note that if the opposite occurred, and we confirmed the previous discrepancy with unitarity, it would have required alteration of current theory to analyse the measured data (e.g. adding more partial waves). This will change the cross section of s and d waves measured at high energies, which is extrapolated to the Gamow window.

2) I think this work may lead to future measurements that will be very significant to the field, but the data presented here are not.

We hope the reviewer agrees that in the first place a confirmation of our completely new method is very “significant to the field”, as is the correction of the historical disagreement with unitarity. No less, before we embark on such a project that all reviewers agree will “lead to future measurements that will be very significant to the field”, we must establish the validity of our completely new method and benchmark it against current world data, as we do here.

3) Again, more comparison with previous data is needed to back up the here presented claims.

See discussion above.

4) The analysis has been very challenging and is excellent.

5) I think the methodology is excellent.

6) The authors give a lot of detail in this paper. It is quite impressive work, but it is not at a level to modify or even question existing claims on the low energy cross section and its impact on astrophysical environments.

Again, we demonstrated that current data measured over the 1^- resonance, where the cross sections are large, fail to show agreement with unitarity. This should lead one to “question existing claims” concerning the low energy data where the cross sections are considerably smaller.

Reviewer #3 (Remarks to the Author):

The carbon/oxygen ratio at the near end of explosive stellar evolution processes has major consequences for the fate of supernovae remnants. The reaction $^{12}\text{C}(\alpha,\gamma)^{16}\text{O}$ is crucial to determine this ratio. Because nuclear physics experiments are difficult, expensive, and take many years of preparation, the progress made to reach the low ^{12}C - α relative motion energies has been daunting. Disentangling the E1 and E2 photon multipolarity component in this reaction has also been plagued by low statistics, near impossibility to reach the required low energies, and by uncertainties in the parameters used in theoretical models (e.g., R-matrix theory), which serve as guide for the experimental data to low energy extrapolation. With the advent of newer and more powerful experimental facilities using real photons (e.g., the ELI facility in Romania), the possibility to explore the inverse reaction, namely, the $\gamma + ^{16}\text{O} \rightarrow \alpha + ^{12}\text{C}$, became feasible. The challenge then becomes the pursue of an accurate detection of the four-momenta (or energies + angular distributions) of final products and the reproduction of their relative motion energy.

This manuscript reports the very first experiment performed in the ELI facility for this purpose. It focus on a novel method to determine the interference angle of the $l = 1$ and 2 partial waves contributing to the reaction. A major accomplishment of this experiment is to clarify a long standing puzzle related to the deviation that some past experiments (based on alternative methods) observed when confronted to robust theoretical predictions based on the unitarity of the scattering matrix (as reported, e.g., in Refs [10,11]).

We thank the reviewer for joining the other reviewer and “fully recommend publication”. We further thank the reviewer for underlining the importance of “confirming robust theoretical predictions based on the unitarity of the scattering matrix”. This long overdue confirmation reverses a historical disagreement with unitarity of measured ϕ_{12} values at 2.0 – 2.6 MeV.

We thank the referee for the thorough review of our paper, we agree with the comments and changed the manuscript accordingly.

The paper is very well written and presents a detailed description of the experimental methods used. Frankly, I cannot tell if there are any flaws in the analysis involving detection efficiency, cross-talks, correlations, background suppression, etc.

This work has immense consequences for the small steps to understand this reaction, with (a) a novel experimental approach, (b) a brand new experimental facility, (c) the resilience and competence of the authors in pursuing this goal for many years (decades) and (d) the new results on the mixing angle, clarifying the validity of known predictions. Because of that, I have no doubt that this work merits publication in this journal. I fully recommend it for publication. However, it is of my opinion that small corrections/addenda would be in place before publication.

The manuscript is well structured and easy to understand by non-experts (a subjective concept). Here are some clarifications that might be necessary.

1- Line 72: "For example, it was concluded that the centre of mass motion of the alpha + ¹²C system leads to large shifts of the centre of mass angles, which were previously ignored [12]." I haven't checked Ref. [12]. But isn't this statement a bit confusing? Is there an obvious reason for that (maybe found in [12])? The authors could add a sentence to explain the reason, if that is easy.

We agree with this reviewer and reviewer 1 who raised the same issue, and we removed this discussion.

2- The authors mention repeatedly the role of the detailed balance theorem, pointing to Eq. (9) of Ref. [17]. Isn't it much better to cite a textbook here? There are many available, some as old as nuclear physics itself.

We agree with this reviewer and on line 151 we included this elementary formula.

3- Line 322: "... and exhibit the correct trend below this energy." Maybe a reference for "correct trend" (Ref. [?]). Or it might be replaced by "predicted trend"?

We agree with this reviewer and reviewer 1 that raised the same issue and we removed this statement.

4- Line 323: "... for the lowest energy angular distribution, we obtained 83 raw counts, not enough to obtain a statistically significant result." Then why there seems to be a smaller error bar at 2.02 MeV than at other energies, for example at 3.28 MeV? Or are they unrelated?

Thank you for highlighting this issue, (as did reviewer 1). For the low statistics data point at the lowest energy, our previous approach of binning the angular data and performing a χ^2 fit of the partial wave decomposition was sub-optimal. At the advice of reviewer 1, we performed a unbinned maximum log likelihood fit to the data. The new fitting method does not significantly change the optimised fit parameters for this energy, but as you highlight, the correct statistical procedure allowed us to extract a more realistic error bar, commensurate with the lower statistics. This is now shown in Fig. 5 of the revised manuscript.

5- Line 312: "... we still use the usual statistical procedure ... to quote the uncertainties of the extracted parameters."

Where in the manuscript are these parameters quoted? Sorry, I could not find them.

To focus our discussion toward the agreement of our measured ϕ_{12} values from 2.0 – 2.6 MeV with unitarity, the optimised fit parameters were not listed explicitly. However, in response to your comment, and a similar point raised by reviewer 1, an electronic supplement to the manuscript has been developed, which includes the angular and cross section data, and the optimised E2/E1 amplitudes and ϕ_{12} values, along with their uncertainties. A discussion of systematic uncertainties is also included.

In place of the quote above, we now state in line 374: "Due to the low statistics (a factor of ~ 5 smaller) of the angular distribution measured at 2.02 MeV we analysed this angular distribution using the unbinned maximum likelihood method."

6- Figure 5 is the main result of this experiment. Here "unitarity" might again deserve a reference.

We added Refs. [12,13] and we thank the reviewer for this comment.

7- I emphasize the complex filtering work has the authors carried out to achieve the final result presented in Fig. 5. But the solid curve also deserves an error band, because as mentioned in line 52, δ_1 and δ_2 are measured. Or is the final effect of their errors too small to show on ϕ_{12} ?

We agree that theoretical uncertainties are just as important, but in this case the elastic scattering is well measured and so are the phase shifts, so the errors are very small on the scale of Fig. 5.

REVIEWER COMMENTS

Reviewer #1 (Remarks to the Author):

As noted in my original report, this paper reports on a significant advance in the experimental determination of the $\sigma^{12}(\text{C}(\alpha, \gamma)^{16}\text{O})$ reaction rate. It is an important development, worthy of publication in a high-profile journal.

Most of my original criticisms have been address effectively in the new version. However, some significant issues remain, particularly regarding the beam energy convolution. I will refer to my original numbering by using parentheses.

1. (1) Here, I think it would be helpful for the authors to explain the difficulty with measurements in the 2.0-2.6 MeV energy range.

Because the cross section here is dominated by the broad E1 resonance, the angular distribution is not very different from a pure E1 angular distribution. Further, φ_{12} is passing through zero in this region. The differential cross section depends on $\cos \varphi_{12}$, which only depends on φ_{12} at second order when φ_{12} is near zero. Measuring φ_{12} thus rests upon subtle changes in the shape of the angular distribution, making it prone to large statistical and systematic uncertainties.

2. (2) Here, the authors have replaced the original Ref. [8], with a different reference, Hammer et al. (2005) [11]. This reference also should not be in the conversation. It is a conference proceedings article with nearly zero information about statistical

procedures, R-matrix parameters, etc... It is disingenuous to discuss deBoer _et al. [8] and Ref. [11] as if they are on equal footing.

There also a lot more data considered in Ref. [8]. I suggest that the authors stop trying to cherry pick low-quality references for making a case for a large uncertainty. This is not the way for the field to make progress.

3. (6) The authors still do not discuss the effect of beam energy uncertainty on their data. This appears to be non-negligible, up to around a 15% effect. This needs to be pointed out.

4. (7) This begs the question of what effect the energy spread has on the cross section extraction. The formalism of Brune and Sayre [NIM A698, 49 (2013)] can be used for this. Specifically, adapting Eqs. (6-8) of that paper:

$$\begin{aligned} \bar{\sigma}(E) &= \frac{\int E \rho(E) \sigma(E) dE}{\int \rho(E) \sigma(E) dE} \\ \bar{\sigma}(E) &= \frac{\int \rho(E) \sigma(E) dE}{\int \rho(E) dE} \sim f \end{aligned}$$

Here, \bar{E} corresponds to the “effective energy” used in the paper under consideration and $\rho(E)$ is the energy distribution of the beam, for which I have assumed a Gaussian with a standard deviation parameter of 300/2.35 keV and truncated at two standard deviations (something like this must result from the photon collimation and/or Q-value cut). The quantity f is the correction factor arising from the the convolution. The assumed $\sigma(E)$ and calculated f factor are shown here:

[image]

First, note that at $E = 2.29$ MeV, we have $f = 0.77$. This indicates that the measured cross section here should be increased by $1/0.77$, i.e., about 30%. The authors need to discuss the effects of convolution and perhaps do a more detailed analysis, as outlined above.

The calculation and plot also indicate very large effects near the $E=2.68$ MeV resonance. How is the measurement at 2.47 MeV not affected by this? This would seem to depend delicately on the beam energy distribution or how Q-value cut is applied. I also wonder why the authors put the 2.64-MeV point on any of their plots? How can this measurement be interpreted? What does the φ_{12} measurement at this energy mean?

The authors need to address these energy convolution issues, including an uncertainty analysis, before I can sign off on publication.

5. (8) It is nice to see the systematic error table for the supplementary information. However, I have some questions. Is the "Range Straggle" error related to the efficiency of the fiducial volume cut? I would think there is there must be some error in the efficiency correction for the fiducial volume cut. Also the beam energy error of 30 keV appears just be giving the percent error in the beam energy. But as noted above, it propagates to a much larger uncertainty in the extracted cross section.

6. (12) I still find Fig. 4 confusing. If the solid curves do not include binning effects, but the data are binned, how can the solid curve be compared the data?

7. (13) For Fig. 4, I suggest not labeling the vertical axis of the figure as “Counts/20^o”. Since an efficiency factor has been applied, the quantity is no longer counts. Perhaps “Yield”?

8. (17) Lines 398-400. I don’t believe the description of “high precision” is apt for the φ_{12} measurements reported in this paper (Fig. 5). The error bars are still large. The only point with a small error bar, 2.64 MeV, is not very meaningful due to the narrow 2^+ resonance at that energy. I suggest something like “significantly improved precision.”

9. (new) I suggest labeling the vertical axis on Fig. 5 as $|\varphi_{12}|$, since the phase defined by $\varphi_{12} = \delta_2 - \delta_1 + \tan^{-1}\eta/2$ actually changes sign near 2.3 MeV. The experiment does not determine the sign since the angular distribution only depends on $\cos \varphi_{12}$.

Reviewer #2 (Remarks to the Author):

I read with great interest the rebuttal arguments by the authors, but I still don’t think that the revised paper provides sufficient new information to justify a publication in Nature. I agree that the reaction is one of the most important ones in Nuclear Astrophysics, I also agree, that a new method might be helpful in providing new information, but this paper only delivers the promise, not the data.

It also adds some misleading statements: “On one hand DeBoer et al. [10] analysed the current world data and concluded that a “level of uncertainty $\sim 10\%$ may be in sight”. However, Hammer et al. [11]

who analysed the very same data, shown in Fig. 5 of Ref. 48 [11], found a larger uncertainty.” DeBoer used many more data sets including different channels and ANC results from indirect studies of the threshold states than Hammer et al. . DeBoer also used with AZURE a largely improved R-matrix code allowing a multichannel analysis, which reduced the uncertainty and in particular removed the possibility of a negative interference term.

The rebuttal demonstrates the insight of the authors in the problems with low energy extrapolation and I agree that there are many uncertainties in the present theoretical extrapolations, which necessitates new data, through indirect transfer or photon induced methods; I agree, that there might be many quantum effects yet unknown or unidentified quantum effects near the threshold, which may make the extrapolation questionable but the paper does not show the direct experimental evidence. It proposes a method and demonstrates that the method might be useful in the future, but so far, it suffers from the same handicap of all direct experimental data, the Coulomb barrier and the lack of count rate at low energies.

This is a very nice instrumentation paper, presenting a good case for applying the technique for the very relevant case of $^{12}\text{C}(\alpha,\gamma)$, but it presents no new information compared to the present evaluation of the reaction rate. I therefore suggest to publish the paper in a more suitable experimental nuclear physics journal.

Reviewer #3 (Remarks to the Author):

My concerns have been addressed in the revision and I am happy to recommend this article for publication in Nature.

Reply to reviewer 1

We thank reviewer 1 for the statement that our paper represents “a significant advance in the experimental determination of the $^{12}\text{C}(\alpha, \gamma)^{16}\text{O}$ reaction rate”, with which we concur. We once again thank reviewer 1 for the latest round of comments, where a number of important points were raised. We agree with the reviewer’s comments, and we address them in the new manuscript as we discuss below. These comments have significantly improved the manuscript.

As noted in my original report, this paper reports on a significant advance in the experimental determination of the $^{12}\text{C}(\alpha, \gamma)^{16}\text{O}$ reaction rate. It is an important development, worthy of publication in a high-profile journal.

Most of my original criticisms have been address effectively in the new version. However, some significant issues remain, particularly regarding the beam energy convolution. I will refer to my original numbering by using parentheses.

1) (1) Here, I think it would be helpful for the authors to explain the difficulty with measurements in the 2.0-2.6 MeV energy range. Because the cross section here is dominated by the broad E1 resonance, the angular distribution is not very different from a pure E1 angular distribution. Further, ϕ_{12} is passing through zero in this region. The differential cross section depends on $\cos \phi_{12}$, which only depends on ϕ_{12} at second order when ϕ_{12} is near zero. Measuring ϕ_{12} thus rests upon subtle changes in the shape of the angular distribution, making it prone to large statistical and systematic uncertainties.

We do not include as much detail as the reviewer suggests here, but we have added a reference [12] to the 2013 Brune and Sayre paper (including figure 12 of that paper), which includes the necessary detail. The manuscript now reads from lines 61 and 138:

“In contrast, in the energy region of $2.0 < E_{\text{cm}} < 2.6$ MeV, both E1/E2 and ϕ_{12} vary rapidly as a consequence of the broad 1^- resonance at 9.58 MeV in ^{16}O . As shown by Brune and Sayre (Fig. 12 of [12]), in this energy region, the variation of ϕ_{12} leads to subtle changes in the measured angular distributions. Therefore, the region of $E_{\text{cm}} = 2.0\text{--}2.6$ MeV, is ideally suited for testing the accuracy of measured angular distributions.

“These fine, detailed differences, which were already highlighted in Ref. [12], allow us to measure ϕ_{12} values that agree with unitarity.”

2) (2) Here, the authors have replaced the original Ref. [8], with a different reference, Hammer et al. (2005) [11]. This reference also should not be in the conversation. It is a conference proceedings article with nearly zero information about statistical procedures, R-matrix parameters, etc. . . . It is disingenuous to discuss deBoer et al. [8] and Ref. [11] as if they are on equal footing. There also a lot more data considered in Ref. [8]. I suggest that the authors stop trying to cherry pick low-quality references for making a case for a large uncertainty. This is not the way for the field to make progress.

We have removed the reference to Hammer et al. and keep the deBoer paper as the best theoretical evaluation of the extrapolation using R-matrix theory. However, we have added the caveat that the data that were utilised by deBoer et al. have previously been shown to have systematic issues that were not considered in the original publications. We briefly highlight

issues with the Stuttgart data and include references to Gai (2013) and Brune and Sayre (2013). The manuscript now reads from line 43:

“In the latest extrapolation to stellar conditions using R-matrix analysis, deBoer et al. [10] examined the current world data and concluded that a “level of uncertainty $\sim 10\%$ may be in sight”. However, Gai [11] and Brune and Sayre [12] revealed significant systematic complications with the data of Assunção et al. [6], and others, that were utilised by deBoer et al..”

3) (6) The authors still do not discuss the effect of beam energy uncertainty on their data. This appears to be non-negligible, up to around a 15% effect. This needs to be pointed out.

For each data point, we now examine the variation of the cross section (average of existing world data) ± 30 keV about the effective centre-of-mass energy. This uncertainty is included in the error bars of Fig. 3 and as a separate column in the electronic supplement. For the data point at 2.64 MeV, close to the narrow 2^+ resonance, this uncertainty is not plotted in Fig. 3 as it extends beyond the scale of the plot (this is noted in the text). A large uncertainty is expected at this point given its proximity to the narrow 2^+ resonance and associated rapid variation of the cross section. We added the following text to the manuscript from line 354:

“Systematic errors in the cross section due to the beam energy uncertainty were estimated by examining the variation of the world data cross section ± 30 keV about each E_{eff} . In Fig. 3, the error bars contain statistical uncertainties (4–15%) and uncertainties due to the beam energy, but exclude the global 11% uncertainty on the beam intensity. The 2.64 MeV point does not show the error due to beam energy uncertainty, as the error bar exceeds the scale of the plot.”

4) (7) This begs the question of what effect the energy spread has on the cross section extraction. The formalism of Brune and Sayre [NIM A698, 49 (2013)] can be used for this. Specifically, adapting Eqs. (6-8) of that paper:

$$\bar{E} = \frac{\int E \rho(E) \sigma(E) dE}{\int \rho(E) \sigma(E) dE}$$

$$f = \frac{\int \rho(E) \sigma(E) dE}{\sigma(\bar{E}) \int \rho(E) dE} .$$

Here, \bar{E} corresponds to the “effective energy” used in the paper under consideration and $\rho(E)$ is the energy distribution of the beam, for which I have assumed a Gaussian with a standard deviation parameter of 300/2.35 keV and truncated at two standard deviations (something like this must result from the photon collimation and/or Q-value cut). The quantity f is the correction factor arising from the convolution. The assumed $\sigma(E)$ and calculated f factor are shown here:

First, note that at $E = 2.29$ MeV, we have $f = 0.77$. This indicates that the measured cross section here should be increased by $1/0.77$, i.e., about 30%. The authors need to discuss the effects of convolution and perhaps do a more detailed analysis, as outlined above.

We thank the reviewer for raising this important point. The reviewer is correct that in order to directly compare our cross sections with the world data in Fig. 3, a correction factor f , as defined above, is required to account for the energy spread of our beam. Using the formalism of Brune and Sayre, along with the average of the current world data for $\sigma(E)$, we have evaluated the relevant correction factors and applied them to our measured cross sections as shown in the updated Fig. 3. The correction factors are listed in the electronic supplement (as the inverse of f). In all instances, this procedure brings our measured cross sections closer to the world data. We added to the manuscript from line 351:

“Using the method described in Ref. [12], a correction factor was applied to the cross sections to account for the energy spread of the gamma beam.”

The calculation and plot also indicate very large effects near the $E=2.68$ MeV resonance. How is the measurement at 2.47 MeV not affected by this? This would seem to depend delicately on the beam energy distribution or how Q-value cut is applied. I also wonder why the authors put the 2.64-MeV point on any of their plots? How can this measurement be interpreted? What does the ϕ_{12} measurement at this energy mean?

We have modified the analysis of the 2.68 MeV angular distribution in response to this point.

We attach a useful figure 1 below, where the signal of total energy deposited in the detector (arb. units), is plotted for candidate ^{16}O events (after the χ^2 cut) in the 2.68 MeV data. There is sufficient resolution to distinguish the broad 1^- and narrow 2^+ states, which are both populated by the gamma beam. By performing a 2 Gaussian fit to the data, we concluded that the narrow 2^+ state contributes to 40% of the overall counts at this energy.

Figure 1: Total energy signal of the detector for candidate ^{16}O events at 2.68 MeV.

In order to extract meaningful information from the angular distribution at this energy, it is now fit with a sum of a pure E2 angular distribution (weighted at 40%) with the partial wave decomposition (60%) of [2]. The updated ϕ_{12} is plotted in Fig. 5 and the new optimised parameters are listed in the electronic supplement.

The following text was added to the manuscript from line 366:

“Due to the 2.64 MeV point’s proximity to the very narrow 2^+ resonance at 2.68 MeV, this angular distribution was fit as a sum of a pure E2 angular distribution (40%) and the partial wave decomposition (60%) [2]. The relative contribution of the narrow E2 resonance was determined by examining the total energy deposited in the detector for all identified ^{16}O events at this beam energy, which was fit with a sum of two Gaussians.”

The influence of the narrow 2^+ resonance on the lower energy 2.47 MeV data point is small. In the following figure 2, we show the total energy deposited in the detector for candidate ^{16}O events at this energy. There is no visible narrow 2^+ contribution as there was in the higher energy data. A single Gaussian fits the data well with a $\chi^2/\nu = 0.84$. A small 4% 2^+ contribution provides the best fit with $\chi^2/\nu = 0.67$. A 10.7% contribution is rejected at the 3σ confidence level.

Figure 2: Total energy signal of the detector for candidate ^{16}O events at 2.47 MeV. Left: best fit with a 4.12% contribution of the narrow 2^+ . Right: fit with 10.7% contribution, rejected with 3σ confidence.

The authors need to address these energy convolution issues, including an uncertainty analysis, before I can sign off on publication.

We hope that reviewer 1 agrees that we have sufficiently addressed these issues in the latest manuscript and that the points raised have improved the analysis.

5) (8) It is nice to see the systematic error table for the supplementary information. However, I have some questions. Is the “Range Straggle” error related to the efficiency of the fiducial volume cut? I would think there is there must be some error in the efficiency correction for the fiducial volume cut. Also the beam energy error of 30 keV appears just be giving the percent error in the beam energy. But as noted above, it propagates to a much larger uncertainty in the extracted cross section.

The range straggle error is related to the efficiency correction. A study was performed, where we examined, through Monte-Carlo simulations, how the uncertainties in the fiducial volume cuts (including range straggle) effect the efficiency correction. A maximum error of 0.6% was determined for the total efficiency. The bin-by-bin angular efficiency correction also has small uncertainties, which are considerably lower than the statistical error bars in the angular distributions. The angular efficiency profiles provided in the electronic supplement now include these uncertainties and the systematic error table has been updated to make the origins of the uncertainties clearer.

We added to the text from line 320:

“Uncertainties in the fiducial volume cuts, including the range straggle of the α and ^{12}C give rise to a $<1\%$ uncertainty in the overall efficiency correction. The event-by-event efficiency curves, and their small systematic uncertainties, are included in the electronic supplement.”

6) (12) I still find Fig. 4 confusing. If the solid curves do not include binning effects, but the data are binned, how can the solid curve be compared the data?

The best fit angular distributions, plotted as solid curves, now contain the effects of angular distribution attenuation factors to account for the binning of the data in fig 4. The text now reads from line 387:

“The solid fit lines in Fig. 4 show the optimised partial wave decompositions convolved with the angular resolution, and include angular distribution attenuation factors (Q_i) to account for angle binning [2].”

7) (13) For Fig. 4, I suggest not labeling the vertical axis of the figure as “Counts/20°”. Since an efficiency factor has been applied, the quantity is no longer counts. Perhaps “Yield”?

We have changed this to label to Yield / 20°.

8) (17) Lines 398-400. I don’t believe the description of “high precision” is apt for the ϕ_{12} measurements reported in this paper (Fig. 5). The error bars are still large. The only point with a small error bar, 2.64 MeV, is not very meaningful due to the narrow 2^+ resonance at that energy. I suggest something like “significantly improved precision.”

We changed the narrative on line 422 to read:

“Our method permitted precision measurements of angular distributions spanning the energy region $E_{cm} = 2.0 - 2.6$ MeV, with significantly improved precision”.

9) (new) I suggest labeling the vertical axis on Fig. 5 as $|\phi_{12}|$, since the phase defined by $\phi_{12} = \delta_2 - \delta_1 + \tan^{-1} \eta/2$ actually changes sign near 2.3 MeV. The experiment does not determine the sign since the angular distribution only depends on $\cos\phi_{12}$.

We have changed the vertical axis label to $|\phi_{12}|$.

Reply to reviewer 2

We thank reviewer 2 for their reconsideration of the manuscript and their latest comments. We attach our responses below and have made new inclusions to our paper in response the points they have raised.

I read with great interest the rebuttal arguments by the authors, but I still don't think that the revised paper provides sufficient new information to justify a publication in Nature. I agree that the reaction is one of the most important ones in Nuclear Astrophysics, I also agree, that a new method might be helpful in providing new information, but this paper only delivers the promise, not the data.

From this response, it seems that reviewer 2 deems only low-energy measurements of this cross section as "sufficient new information". We are confident that before attempting to extrapolate to stellar energies, one must consider the quality of the data used in the extrapolation. Here we highlight, using an entirely new experimental approach, a key flaw in the current world data, many of which are incongruent with the fundamental prediction of unitarity. As we discuss below, we demonstrate that many data used to extrapolate to stellar conditions, for example, in Ref. [10], disagree with the prediction of unitarity. In particular, all ten low energy data points of Assunção et al., down to 1.31 MeV, disagree with this fundamental prediction, highlighting significant systematic issues in the measured angular distributions. This requires re-assessment of the current world data and their extrapolation to stellar conditions.

In the attached figure below, we once again compare our data for ϕ_{12} , shown in Fig. 5 of the manuscript, with the world data on ϕ_{12} .

E1-E2 mixing phase (ϕ_{12})

We emphasise that the yellow framed figure was prepared by Professor Wolfgang Hammer for his retirement address. Namely, Professor Hammer proudly displayed his lifetime contribution to the field and the contribution of his lab at Stuttgart, that he directed for several decades. We note that Professor Hammer labelled the theoretical curve, shown by the green line, as R-Matrix. He shows major deviations of his (EUROGAM) results from the theoretical prediction.

He also showed that other measurements did not do much better and also disagreed with theory. This would be of little consequence if the prediction was just R-Matrix theory, except that unbeknownst to Professor Hammer, and as we discuss in our paper from line 82, the green theory line is in fact a fundamental prediction based on quantum mechanics itself. Professor Hammer therefore showed a significant conflict of the world data with quantum mechanics.

We are confident that reviewer 2 would agree that as we state in line 90, a disagreement with quantum mechanics “cannot be overlooked”. Indeed, as we discuss in line from line 70, for the last 47 years, ever since Dyer and Barns 1974 seminal paper [2], the majority of measurements of ϕ_{12} disagree with the prediction of quantum mechanics over the region of 2.0 – 2.6 MeV. This major oversight has now been reprimanded by us. We demonstrate that data must and do agree with quantum mechanics.

As we now conclude in line 431, we have developed in our paper a major tool to judge the quality of data; one must demand that measured angular distributions agree with quantum mechanics. A disagreement with quantum mechanics, must be considered a sign of poorly understood systematic errors.

Specifically, we show the modern data of the Stuttgart group in the figure below. All ten data points down to 1.3 MeV disagree with the prediction of quantum mechanics (blue line). However, these are the data that are used, for example, by deBoer *et al.* [10] to extrapolate to astrophysical energies. We are confident that reviewer 2 would agree that we have revealed a major issue that has hitherto remained unresolved, and one should now question extrapolations based on these and similar data, in light of our research.

It also adds some misleading statements: “On one hand DeBoer et al. [10] analysed the current world data and concluded that a “level of uncertainty $\sim 10\%$ may be in sight”. However, Hammer et al. [11] who analysed the very same data, shown in Fig. 5 of Ref. 48 [11], found a larger uncertainty.” DeBoer used many more data sets including different channels and ANC results from indirect studies of the threshold states than Hammer et al. DeBoer also used with

AZURE a largely improved R-matrix code allowing a multichannel analysis, which reduced the uncertainty and in particular removed the possibility of a negative interference term.

We agree with reviewer 2 and we replaced the narrative with the following statement from line 43:

“In the latest extrapolation to stellar conditions using R-matrix analysis, deBoer et al. [10] examined the current world data and concluded that a “level of uncertainty $\sim 10\%$ may be in sight”. However, Gai [11] and Brune and Sayre [12] revealed significant systematic complications with the data of Assunção et al. [6], and others, that were utilised by deBoer et al..”

The rebuttal demonstrates the insight of the authors in the problems with low energy extrapolation and I agree that there are many uncertainties in the present theoretical extrapolations, which necessitates new data, through indirect transfer or photon induced methods; I agree, that there might be many quantum effects yet unknown or unidentified quantum effects near the threshold, which may make the extrapolation questionable but the paper does not show the direct experimental evidence. It proposes a method and demonstrates that the method might be useful in the future, but so far, it suffers from the same handicap of all direct experimental data, the Coulomb barrier and the lack of count rate at low energies.

The majority of our data that confirm the prediction of quantum mechanics, shown in Fig. 4, (except for the one data point at 2.02 MeV) do not lack statistics, and we demonstrate extraction of the E1-E2 phase with sufficient accuracy in Fig. 5.

This is a very nice instrumentation paper, presenting a good case for applying the technique for the very relevant case of $^{12}\text{C}(\alpha,\gamma)$, but it presents no new information compared to the present evaluation of the reaction rate. I therefore suggest to publish the paper in a more suitable experimental nuclear physics journal.

Nature Communications (and indeed the Nature Portfolio of journals) has a section devoted to “Techniques and Instrumentation” (<https://www.nature.com/subjects/techniques-and-instrumentation/ncomms>). Besides, we already published several instrumentation papers [16, 21] that describe our new technique. In this paper we present significant results. We show, for the first time in 47 years, data that agree with quantum mechanics. This is not an “instrumentation paper”. Our result will resonate with scientists who consider the historical disagreement with quantum mechanics a major issue that must be reprimanded, as we do here for the first time.

We would finally like to address the comment: “it presents no new information compared to the present evaluation of the reaction rate”. We concur that the goal of measurements in nuclear astrophysics is to extrapolate stellar conditions from terrestrial measurements. It is exactly for this reason that we devised a test of the data used in these extrapolations. We are confident the reviewer agrees that requiring data to agree with quantum mechanics is self-evident.

Our initial manuscript focussed on the deviation of measurements of ϕ_{12} from the prediction of unitarity. Reviewer 2 has since highlighted that scientists in our field are particularly interested in the impact that this discrepancy has on extrapolations astrophysical energies, and we thank the reviewer for raising this point. The current manuscript more clearly highlights significant systematic issues with the global data that are used to extrapolate to astrophysical

energies. The E1 and E2 cross sections of the world data, used in the extrapolations, sensitively depend on the shape of the angular distributions. We have demonstrated these must have incorrect shapes, given that their extracted ϕ_{12} values disagree with the predictions of quantum theory. Therefore, it is reasonable to doubt the extracted E1 and E2 cross sections used in the extrapolations to stellar conditions.

REVIEWER COMMENTS

Reviewer #1 (Remarks to the Author):

This paper reports on a significant new experimental approach to the $^{12}\text{C}(\alpha,\gamma)^{16}\text{O}$ reaction. This paper provides new and precise angular distribution data. I feel that the new technique is very promising a worthy of publication in high-profile journal. However, it is also true that this work will have very little impact on the evaluation of the reaction rate at stellar temperatures. In my view the technique and demonstration of accurate measurements at higher energies justify a Nature publication.

The majority of my criticisms and suggestions have been reasonable implemented. I have the following comments:

1) Line 47. The new sentence "However, Gai [11] and Brune and Sayre [12] revealed significant systematic complications with the data of Assunção et al. [6], and others, that were utilized by deBoer et al." is misleading. It suggests that the conclusions of Ref. [10] are invalid, due to the work Refs. [11,12]. But this is not the case. Systematic uncertainties in the input data, in light of Refs. [11,12], were fully considered in Ref. [10].

2) Line 84. "... $\phi_{[12]}$ [2]..." I believe the [2] should be [2,13] to be consistent with the earlier text. Also, [13] includes a derivation which is very general, in the context of R-matrix (multi-level plus external capture).

3) Line 103. "..., to include more partial waves,..." Can the authors provide a reference for this? Otherwise I suggest removing it.

4) Lines 105-109. Rather than "judging data" I would say "identifying systematic errors". I'm not sure what one does with a "judgment". All experiments have systematic uncertainties.

5) Line 360. The text says "The 2.64 MeV point does not show the error due to beam energy uncertainty, as the error bar exceeds the scale of the plot." I suggest not plotting the point at all in this case, as it is otherwise misleading.

6) Lines 432-438. The same comments in 4) apply here. In addition, the sentence "Hence, we highlight significant issues with current data used to extrapolate to stellar conditions." is an overstatement. Modern R-matrix analyses, such as deBoer et al., simultaneously fit the gamma-ray angular distribution data and the elastic scattering, and are not significantly effected by this issue. It was a more significant problem in the past, when ϕ_{12} was treated as a free fit parameter.

Reviewer #2 (Remarks to the Author):

I have not changed my mind, I think the paper presents an interesting alternative method, but it is neither more precise than previous attempts nor does it present any significantly new data that would affect the present rate of $^{12}\text{C}(\alpha, \text{g})$. I think it should be published as complementary data set, but I don't think it carries the scientific relevance that would justify its publication in Nature or in PRL for that matter.

Reply to reviewers

Reviewer 1

This paper reports on a significant new experimental approach to the $^{12}\text{C}(\alpha,\gamma)^{16}\text{O}$ reaction. This paper provides new and precise angular distribution data. I feel that the new technique is very promising and worthy of publication in high-profile journal. However, it is also true that this work will have very little impact on the evaluation of the reaction rate at stellar temperatures. In my view the technique and demonstration of accurate measurements at higher energies justify a Nature publication.

The majority of my criticisms and suggestions have been reasonable implemented. I have the following comments:

We thank reviewer 1 for their exemplary work in reviewing our paper. We found all the comments fair and precise and made suitable changes to the manuscript as detailed below.

1) Line 47. The new sentence "However, Gai [11] and Brune and Sayre [12] revealed significant systematic complications with the data of Assunção et al. [6], and others, that were utilized by deBoer et al." is misleading. It suggests that the conclusions of Ref. [10] are invalid, due to the work Refs. [11,12]. But this is not the case. Systematic uncertainties in the input data, in light of Refs. [11,12], were fully considered in Ref. [10].

We agree and we removed the entire sentence starting in line 47. We did not intend to mislead the reader.

2) Line 84. "... ϕ_{12} [2]..." I believe the [2] should be [2,13] to be consistent with the earlier text. Also, [13] includes a derivation which is very general, in the context of R-matrix (multi-level plus external capture).

We agree and have added the reference in line 51.

3) Line 103. "..., to include more partial waves,..." Can the authors provide a reference for this? Otherwise I suggest removing it.

There was an attempt by a UConn graduate student to reprimand the disagreement with unitarity by adding the $l = 3$ partial wave, and Professor Gai exchanged such comments with Professor Wiescher (private communication), but these are not worthy of referral. We have removed this statement.

4) Lines 105-109. Rather than "judging data" I would say "identifying systematic errors". I'm not sure what one does with a "judgment". All experiments have systematic uncertainties.

We agree and we changed the narrative as suggested on line 103.

5) Line 360. The text says "The 2.64 MeV point does not show the error due to beam energy uncertainty, as the error bar exceeds the scale of the plot." I suggest not plotting the point at all in this case, as it is otherwise misleading.

We agree and have removed this data-point from Fig. 3 and removed the line in the narrative.

6) Lines 432-438. The same comments in 4) apply here. In addition, the sentence "Hence, we highlight significant issues with current data used to extrapolate to stellar conditions." is an overstatement. Modern R-matrix analyses, such as deBoer et al., simultaneously fit the gamma-ray angular distribution data and the elastic scattering, and are not significantly effected by this issue. It was a more significant problem in the past, when ϕ_{12} was treated as a free fit parameter.

We replaced the narrative from "to judge the quality" with "to evaluate systematic errors" and we removed the sentence starting with "Hence we highlighted..." in the previous draft.

Reviewer 2

I have not changed my mind, I think the paper presents an interesting alternative method, but it is neither more precise than previous attempts nor does it present any significantly new data that would affect the present rate of $^{12}\text{C}(\alpha, \gamma)$. I think it should be published as complementary data set, but I don't think it carries the scientific relevance that would justify its publication in Nature or in PRL for that matter.

We thank reviewer 2 for reading the latest manuscript. We are surprised by the statement that our data are not "more precise than previous attempts". Our data show, for the first time over the 1^- resonance region, a precise agreement with the prediction of quantum mechanics – a fit not achieved over the last 47 years by any other measurement.

By their own admission, reviewer 2 stated that "the group is pioneering a completely different way of measuring the $^{12}\text{C}(\alpha, \gamma)$ reaction with completely different systematic uncertainties." Our pioneering new technique with different systematic uncertainties is the very reason that we are able to measure accurate angular distributions. In doing so, we reprimand the historical disagreement of the world data with a fundamental prediction of quantum mechanics, which is a highly significant result.

Additionally, the E1 and E2 cross sections at lower energies are very sensitive to the value of ϕ_{12} as demonstrated by table I in Assunção 2006. In that work, the E2 cross section is seen to vary by over 200% when comparing the fits where ϕ_{12} is a free parameter and where ϕ_{12} is fixed to the value predicted by unitarity. Our paper highlights major systematic issues in measuring accurate and correct angular distributions and this should be a significant consideration when evaluating current data and when planning further measurements of this important cross section at low energies.

REVIEWERS' COMMENTS

Reviewer #1 (Remarks to the Author):

My previous opinion regarding the suitability of this work for publication in Nature remains unchanged. This version satisfactorily address all of my previous criticisms and suggestions.

25th August 2021

Communication to reviewer 1

We thank reviewer 1 for their report that was sent to us on June 29th after which the editors informed us that “In light of their advice I am delighted to say that we are happy, in principle, to publish a suitably revised version in Nature Communications”. Once again, we thank reviewer 1 for their exemplary review.

Meanwhile, when preparing the analyses of more data that we have “on tape” we recognised a small oversight in the extraction of the in-plane angle α , which stems from “over analysing” our data as we discuss below. This required a thorough review of the data analyses, hence the long delay. This has resulted in an update to the results shown earlier in Fig. 5. We present the new results in Fig. 5 of the resubmitted manuscript, along with a revised narrative (shown in blue). The agreement of the extracted ϕ_{12} values with the theoretical prediction of unitarity still follow the correct trend with a χ^2/DoF of 0.65.

We feel that the strength of our paper remains as stated earlier by this reviewer: “This paper reports the first results using a very new and different technique. Considering the likely improvements in the detector, beam intensity, beam time, data analysis, etc..., it is very probable that this technique will in the future yield results that surpass the quality of the currently available data. In this sense, this paper is an important development and represents a significant milestone. In my view, this latter point makes these results noteworthy and valuable to publish”.

Indeed, all three reviewers emphasised the value of our entirely new technique. We believe that despite the minor changes to the result regarding the theoretical prediction of the $E1-E2$ phase angle, we still demonstrate the strength of our method.

We now explain the oversight in our data analyses:

We stated in our earlier version of the manuscript on line 480: “From each image, the angle of the track in the y-z plane, relative to the beam direction, α , was extracted. The y-z coordinates of each pixel in a track were plotted and a linear fit was performed. The fit was weighted by each pixel's intensity.”

We intended to extract the in-plane angle α with the best accuracy, hence the fit was weighted by each pixel's intensity. However, since the window of our gated MCP image intensifier, shown in Fig. 1, is made from optical fibers, the light arrives at the CCD camera in buckets and is not smoothly distributed over the pixels of the track. We subsequently found that for some events, pixels with large intensities can dominate the fit and lead to an incorrectly extracted angle α . We

changed our fitting algorithm to remove this weight factor, which leads to the slightly modified angular distributions shown in the new Fig. 4.